# NAA and 6-BA promote accumulation of oleanolic acid by JA regulation in *Achyranthes bidentata* Bl

**Yanqing Liu**[1], **Li Tang**[1], **Can Wang**[1], **Jinting Li**[1,2]*

**1** College of Life Sciences, Henan Normal University, Xinxiang, China, **2** Engineering Laboratory of Biotechnology for Green Medicinal Plant of Henan Province, Xinxiang, China

* Ljt66882004@126.com

**Data Availability Statement:** All the readings obtained in this study have been uploaded to the NCBI Sequence Read Archive, and the accession number is PRJNA350183. All other relevant data

## Abstract

Application of plant growth regulators has become one of the most important means of improving yield and quality of medicinal plants. To understand the molecular basis of phyto-hormone-regulated oleanolic acid metabolism, RNA-seq was used to analyze global gene expression in *Achyranthes bidentata* treated with 2.0 mg/L 1-naphthaleneacetic acid (NAA) and 1.0 mg/L 6-benzyladenine (6-BA). Compared with untreated controls, the expression levels of 20,896 genes were significantly altered with phytohormone treatment. We found that 13071 (62.5%) unigenes were up-regulated, and a lot of differentially expressed genes involved in hormone or terpenoid biosynthesis, or transcription factors were significantly up-regulated. These results suggest that oleanolic acid biosynthesis induced by NAA and 6-BA occurs due to the expression of key genes involved in jasmonic acid signal transduction. This study is the first to analyze the production and hormonal regulation of medicinal *A. bidentata* metabolites at the molecular level. The results herein contribute to a better under-standing of the regulation of oleanane-type triterpenoid saponins accumulation and define strategies to improve the yield of these useful metabolites.

## Introduction

*Achyranthes bidentata* Blume, a member of the family Amaranthaceae, is a well-known and widely prescribed traditional Chinese herb that has been listed in the Chinese Pharmacopoeia [1]. In China, this species is primarily distributed in the Guhuaiqingfu area, located in Jiaozuo in Henan Province. *A. bidentata* is also called 'Huainiuxi'. Its dried root is an important herbal medicine that is used to maintain liver and kidney function, strengthen the muscles and bones, promote blood flow, remove blood stasis, and increase longevity [2–4]. Oleanane-type triterpenoid saponins belong to the major active phytochemical compounds in *A. bidentata*, and possess hepatoprotective effects as well as anti-inflammatory, antioxidant, and anticancer activities [5, 6].

Oleanolic acid, a pentacyclic triterpenoid, is an isoprenoid-derived compound. The triter-penoid biosynthetic pathway has been found in many plants, such as *Platycodon grandiflorum*

are within the manuscript and its Supporting Information files.

**Funding:** The National Nature Science Foundation of China (81274076) and the Key Projects of Henan Province Colleges and Universities (17A180026).

**Competing interests:** The authors have declared that no competing interests exist.

[7], *Panax japonicus* [8, 9], *Astragalus membranaceus Bge.* [10], *Anemone flaccida* [11], *Phyllanthus amarus* [12], *Gynostemma pentaphyllum* [13], *Panax ginseng* [14], *Eleutherococcus senticosus* [15], and *A. bidentata* [16]. In general, the triterpenoid biosynthetic pathway can be divided into three steps: synthesis of universal terpenoid precursors, formation of carbon skeletons, and modification of triterpenoid skeletons. Triterpenoid precursors are synthesized mainly through the cytoplasmic mevalonate (MVA) pathway. The plastidic 2-C-methyl-D-erythritol-4-phosphate (MEP) pathway is believed to be a minor pathway for triterpenoid biosynthesis [17]. Triterpenoid skeleton synthesis genes include *FPS* (farnesyl diphosphate synthase), *SS* (squalene synthase), *SE* (squalene epoxidase), and *OSCs* (oxidosqualene cyclases). The cyclization of 2,3-oxidosqualene catalyzed by OSCs is a key step in the biosynthesis of triterpenoid saponins and sterols. β-amyrin synthase (*β-AS*) and cycloartenol synthase (*CAS*) catalyze the conversion of 2,3-oxidosqualene to β-amyrin and cycloartenol, which are precursors for oleanolic acid and phytosterols, respectively. A lot is known about several genes involved in modification of the triterpenoid skeleton downstream of the cyclization step are known. Some cytochrome P450 monooxygenases (CYP450s) and glycosyl transferases (GTs) were shown to catalyze modifications of triterpenoid skeletons, including hydroxylation and glycosidation. An increasing number of studies have indicated that the CYP716 family belongs to the CYP85 clan of CYP450s, and is involved in biosynthesis of various triterpenoids including oleanane backbones [18, 19]. For example, CYP716A12 is involved in oxidation of the β-amyrin skeleton, which modifies the oleanane backbone [20]. CYP716A52v2 catalyzes the conversion of β-amyrin to oleanolic acid in *P. ginseng* [21]. However, the CYP450 genes involved in the synthesis of oleanolic acid in *A. bidentata* are unknown.

Application of plant growth regulators has become one of the most important means of improving yield and quality of medicinal plants. Numerous studies have shown that plant growth regulators not only regulate the growth of medicinal plants, but also secondary metabolism, affecting the synthesis of phenols, terpenoids, nitrogen compounds, and other medicinally active ingredients [22]. For example, in *P. quinquefolium* adventitious root, treatment with methyl jasmonate (MeJA) results in an increase in ginsenoside content (43.66 mg/g compared to 8.32 mg/g in control group) [23]. In *P. ginseng* hairy root cultures, the three phytohormones, 2,4-dichlorophenoxyacetic acid (2,4-D), NAA, and indolebutyric acid (IBA), improved the growth of the hairy roots and promoted saponin accumulation to different degrees, and 0.5 mg/L IBA significantly promoted the accumulation of total saponins and ginsenoside Rb1 [24]. In *Calendula officinalis* in vitro hairy root culture, jasmonic acid (JA) treatment increased both the accumulation of oleanolic acid saponins in the hairy root tissue (up to 20-fold) and, in particular, the secretion of these compounds to the medium (up to 113-fold) [25]. In addition, in previous studies we found that application of 1.0 mg/L IBA alone and in combination with 1.0 mg/L 6-BA was advantageous to growth and increased total oleanolic acid and ecdysterone contents in *A. bidentata* roots. Soaking the seeds of *A. bidentata* with 0.15 mmol/L MeJA can increase the accumulation of oleanolic acid in its roots and leaves, which are significantly increased by 114.3% and 60% respectively compared with the control group [26, 27]. Although multiple studies have shown that combining cytokinin and auxin promotes secondary metabolism in medicinal plants, the underlying molecular signaling mechanisms are largely unknown, except for JA. In the absence of JA-Ile, the JAZ repressor proteins bind to the MYC2 transcription factor, thereby blocking downstream JA response. In the presence of JA-Ile, COI1 binds to the JAZ proteins, which leads to their ubiquination by the SCF$^{COI1}$ complex and their subsequent degradation by the 26S proteasome. Degradation of the JAZ proteins releases MYC2, leading to transcriptional activation of the JA-responsive genes. The JAZ proteins contain a conserved TIFY motif within the ZIM domain that mediates homo- and hetero-dimeric interactions between different JAZ proteins. The ZIM domain also functions to

recruit transcriptional corepressors through the novel interactor of JAZ (NINJA) protein. The JAZ proteins contain a Jas domain that is required for the interaction of both COI1 and a broad array of TFs. In the presence of JA or its bioactive derivatives, JAZ proteins are degraded and freeing TFs for expression of specific sets of JA-responsive genes, regulate enzymes involved in the biosynthesis of secondary metabolites, such as ginsenoside, artemisinin, and vinblastine [28].

We have previously conducted extensive studies on the cultivation, phytochemistry, and pharmacology of *A. bidentata* and the relationship between plant structure and accumulation of active components. However, the molecular mechanisms of phytohormone-elicited plant growth and secondary metabolism are still under investigation, largely due to the lack of genomic information [29, 30]. In the present study, RNA-seq was used to sequence the transcriptomes of young *A. bidentata* leaves treated with a combination of 2.0 mg/L NAA and 1.0 mg/L 6-BA and compared to untreated controls with the aim of uncovering the molecular mechanisms of enhanced oleanolic acid production following the application of exogenous hormones.

## Materials and methods

### Plant materials and treatment

Seeds of *A. bidentata* were planted and grown in the natural environment at the Wenxian Agricultural Science Institute of Henan Province, China. 20 days after emerging, the leaves were sprayed with a mixture of different concentrations of NAA and 6-BA until there was liquid dripping at the edge of the blade. The mixture included: 1.0 mg/L NAA + 0.5 mg/L 6-BA (T1); 2.0 mg/L NAA + 1.0 mg/L 6-BA (T2); 4.0 mg/L NAA + 2.0 mg/L 6-BA (T3) and 8.0 mg/L NAA + 4.0 mg/L 6-BA (T4). Experimental controls were treated at the same time with equal volumes of distilled water. At various times after treatment, *A. bidentata* plants were collected to measure growth parameters. Roots, which are usually used for medicinal purposes, were collected to determine oleanolic acid content, and leaves were sampled to measure chlorophyll (Chl. a, Chl. b) and carotenoid contents, endogenous JA content, and to extract RNA for RNA-seq and quantitative real-time PCR (qRT-PCR) as described below. All experiments were performed independently in triplicate.

### Determination of growth indexes and oleanolic acid content

Twenty days after treatment, *A. bidentata* plants from all groups were collected to measure growth parameters, including root length, fresh weight of root, dry weight of root, plant height, fresh weight of plant, and dry weight of plant. In addition, the dried roots from all groups were ground into powder to measure oleanolic acid content. Extraction and quantification of oleanolic acid by high-performance liquid chromatography (HPLC) were conducted according to previously published methods [30]. We weighed 0.5 g dry weight of *A. bidentata* roots powder into a triangle bottle (repeated 3 times for each sample group), added 10 mL of methanol to it, sonicated for 40 minutes, and then concentrated using a rotary vacuum evaporator to completely dry the sample. Then 10 mL of 4 mol / L hydrochloric acid was added, and the extract was hydrolyzed at 85°C for 1 h. After cooling, 10 mL of chloroform was added to the sample, and then two reflux extractions were performed at 60°C for 15 minutes each. The lower liquid was collected and concentrated to dryness under reduced pressure. The residue was dissolved in 3 mL of methanol, filtered through a filter (0.22 μm), and measured by HPLC. The assay conditions were mobile phase: methanol-water-glacial acetic acid (90:10:0.1); flow rate: 0.9 mL/min; column temperature: 25°C; detection wavelength: 210 nm.

## Determination of chlorophyll (Chl. a, Chl. b) and carotenoid contents

The second pair of leaves from the top of plants (Control and T2 group) was harvested to measure the contents of chlorophyll *a*, *b*, and carotenoid. Fresh chopped leaves (0.1 g) added in 10 mL of 80% acetone (v/v) at room temperature in a dark environment for 24 h, and then measured the absorbance at 645 nm, 663 nm and 470 nm, respectively, for detecting chlorophyll a, b and carotenoids contents [31].

## Assay of endogenous JA content

The leaves of the Control and T2 groups were collected at 0, 6, 12, 24, 48, and 72 h to measure endogenous JA content using HPLC. Extractions of endogenous JA were done as described previously [32].

## Library construction and high-throughput RNA sequencing

Young and healthy *A. bidentata* leaves (2nd and 3rd leaves from the top) were treated with 2 mg/L NAA and 1 mg/L 6-BA or distilled water (as Control) and harvested separately at 0, 3, 6, and 9 days after treatment. Total RNA was extracted from each leaf sample for RNA-seq. Libraries were constructed from total RNA isolated from the Control and treated leaves. Equal amounts of total RNA extracted from samples taken at different time points were pooled together. The methods for total RNA extraction and cDNA library construction were described previously [16]. Three biological replicates were used for RNA extraction and two replicates were used for leaf transcriptome sequencing. The libraries were sequenced on the Illumina HiSeq 2500 platform. All reads generated in this study are available from the NCBI Sequence Read Archive database (http://www.ncbi.nlm.nih.gov/sra/) under the project accession number PRJNA350183.

## RNA sequencing data analysis

The raw RNA-seq reads were processed to remove sequencing adapters and low quality bases using Trimmomatic [33], and clean reads shorter than 80 bp were discarded. Then, the high-quality reads were assembled into unigenes with previously published data [16] using Trinity [34] with "min_kmer_cov" set to 10. The unigenes were compared to the ribosome RNA database [35] and GenBank Nucleotide (nt) database using BLAST+ [36] to remove ribosome RNA, viral and bacterial DNA, and other contaminants. To remove redundancies in the cleaned unigenes, they were further *de novo* assembled using iAssembler [37] with 97% minimum percent identify. The high-quality clean reads were aligned back to non-redundant unigenes using Bowtie [38], allowing up to 2 mismatches. Reads per kilobase of exon model per million mapped reads (RPKM) was calculated to determine the expression levels of unigenes. The unigenes with extremely low expression (RPKM < 0.001) in all samples were discarded in downstream analysis.

The functional descriptions of unigenes were predicted using automated assignment of human readable descriptions (AHRD: https://github.com/groupschoof/AHRD). The unigenes were queried against the UniProt database using BLAST+ [36], and the Gene ontology (GO) terms were assigned to the unigenes based on the GO terms annotated to their corresponding homologs in the UniProt database. Biochemical pathways were predicted based on the AHRD and GO annotation of unigenes using the Pathway Tools program [39].

Statistical analysis was performed using DESeq [40] and those unigenes with fold changes ≥2 or ≤0.5 and with adjusted $P < 0.05$ were categorized as differentially expressed genes (DEGs). GO and pathway enrichment analysis was done to gain insight into the biological function of

DEGs. GO enrichment analysis of DEGs was implemented in GO::TermFinder [41]. The pathway enrichment analysis was performed using the in-house Perl scripts. The raw *P* values generated by GO and pathway enrichment analysis were corrected using Benjamini-Hochberg methods [42] and the corrected P < 0.05 were considered significantly enriched in DEGs.

### Phylogenetic analysis

Phylogenetic analysis was performed based on deduced amino acid sequences of CYP450 from *A. bidentata* and other plants. All of the deduced amino acid sequences were aligned with Clustal W with a delay divergent cutoff of 30%, and the other parameters set to defaults as described previously [43]. The evolutionary distances were computed using the Jones-Taylor-Thornton (JTT) method. For the phylogenetic analysis, a neighbor-joining tree was constructed with bootstrap values obtained after 1000 replications using MEGA7.0 [44].

### Real-time quantitative reverse transcription polymerase chain reaction (qRT-PCR) validation and expression analysis

Six DEGs involved in triterpenoid saponin metabolic pathways were chosen for real-time qRCR verification of the Illumina RNA-seq results. The gene-specific primers, designed using Primer Premier 5.0 software, are listed in S1 Table. All reactions were performed in 96-well plates in a LightCycler 96 (Roche, Switzerland) using SYBR®Green Master Mix (Vazyme, China) according to the manufacturer's instructions. All reactions were performed for three biological replicates with three technical replicates per experiment, and for each sample the results were expressed relative to the expression levels of an internal reference gene, *Actin* (UN011760) using the $2^{-\Delta\Delta Ct}$ method.

### Data analysis

Results are presented as means ± standard deviations. All data were subjected to one-way analysis of variance (ANOVA), and means were compared with Student's t-test at a 5% or 1% level of probability using SPSS 20.0.

## Results

### Effects of NAA and 6-BA on the growth of *A. bidentata*

The growth parameters of the plants 20 days after treatment with various combinations of NAA and 6-BA are presented in Fig 1. Lower concentrations of NAA and 6-BA (treatment T1 and T2) improved these parameters, while higher concentrations (particularly T4) exerted adverse effect. Under T2 treatment, all morphological parameters were at the highest values, and root length, fresh weight of root, dry weight of root, plant height, fresh weight of plant, and dry weight of plant were significantly increased by 19.9%, 39.5%, 35.6%, 15.9%, 42.4%, and 19.2%, respectively, compared to the Control (P < 0.05). Chlorophyll *a*, chlorophyll *b* and carotenoid contents in T2 group leaves were measured, and compared with the Control group, all were significantly decreased (S2 Table). The results indicated that 2.0 mg/L NAA and 1.0 mg/L 6-BA treatment significantly reduced biosynthesis of photosynthetic pigments.

### Effects of NAA and 6-BA on oleanolic acid content in *A. bidentata* roots

In this study, the roots of *A. bidentata* were collected 20 days after treatment to measure the levels of oleanolic acid. The oleanolic acid content in all treatment groups except T4 was significantly higher than in Control roots; the content in T1 was the highest, followed by T2, and oleanolic acid contents in T1 and T2 were not significantly different (Fig 2). These results

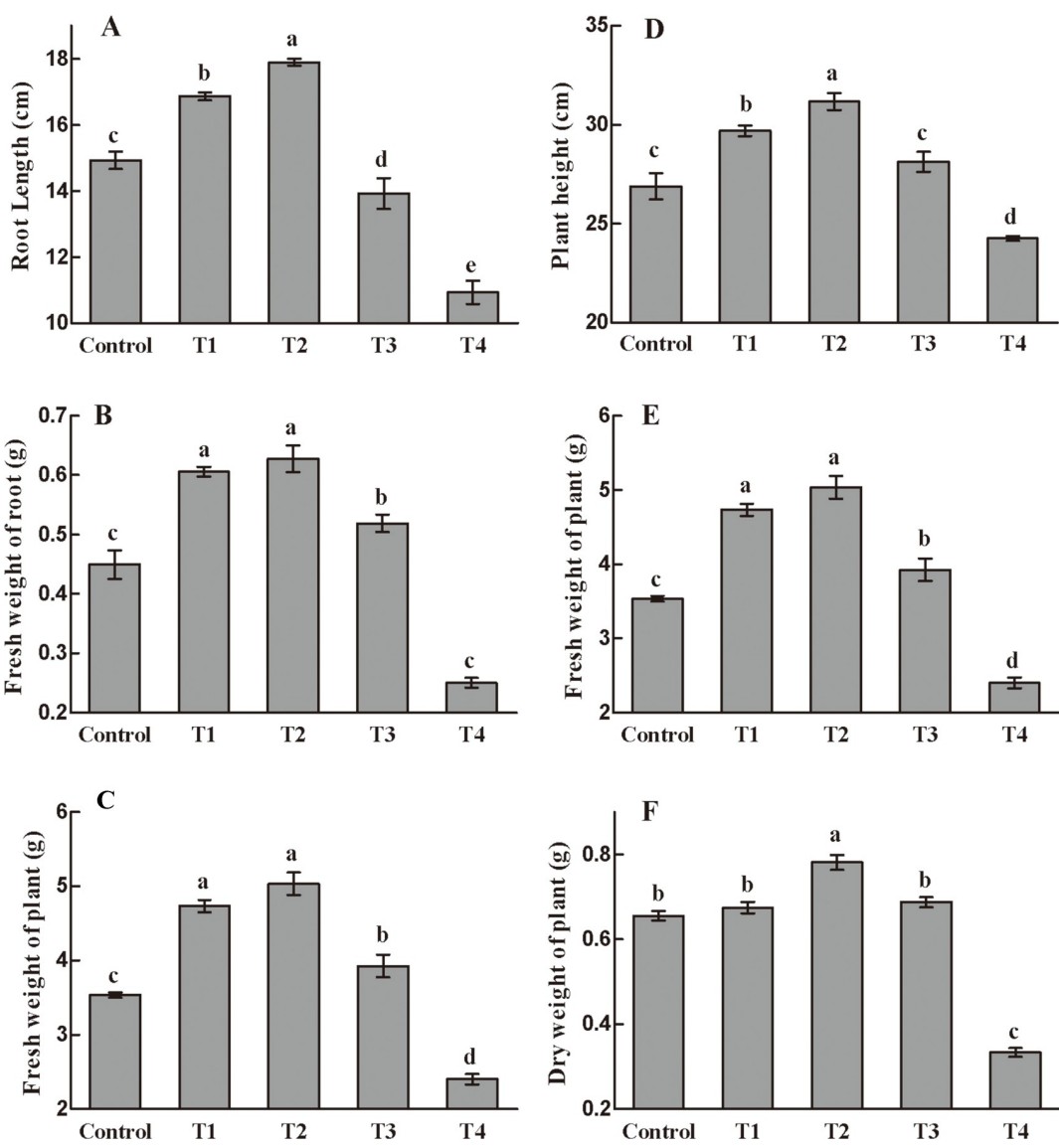

**Fig 1. Effects of NAA and 6-BA on the growth of *A. bidentata* leaves.** A. Root length, B. fresh weight of root, C. dry weight of root, D. plant height, E. fresh weight of plant, and F. dry weight of plant. T1: 1.0 mg/L NAA + 0.5 mg/L 6-BA; T2: 2.0 mg/L NAA + 1.0 mg/L 6-BA; T3: 4.0 mg/L NAA + 2.0 mg/L 6-BA; T4: 8.0 mg/L NAA + 4.0 mg/L 6-BA.

indicated that low concentrations of NAA and 6-BA combinations significantly promoted the accumulation of oleanolic acid, while high concentrations inhibited accumulation. Therefore, the T2 group was taken as the focus of our investigation and analysis in this study. Furthermore, to determine the molecular mechanism of auxin and cytokinin-induced changes in oleanolic acid accumulation, we used Control and T2 (2.0 mg/L NAA and 1.0 mg/L 6-BA treatment) *A. bidentata* leaves as experimental materials for constructing Control and treated (T) RNA-seq libraries, respectively.

## RNA-seq analysis of *A. bidentata* leaves

The transcriptomes of Control and T leaves were sequenced in order to gain a comprehensive overview of the transcriptional response of *A. bidentata* to NAA and 6-BA. To enhance data

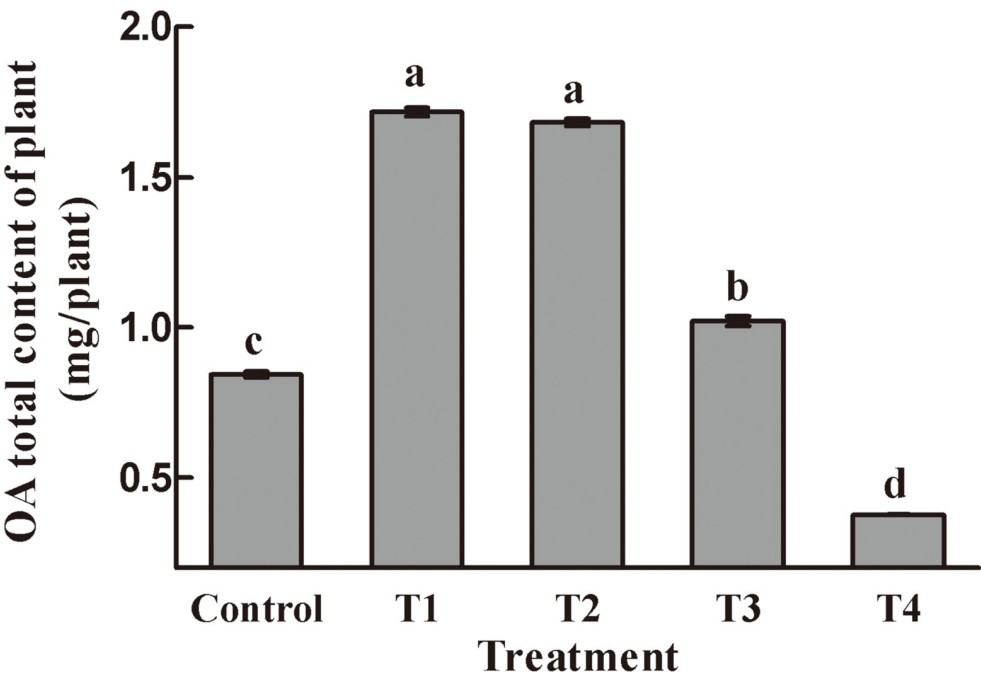

**Fig 2. Effects of NAA and 6-BA on oleanolic acid content in *A. bidentata* roots.** T1: 1.0 mg/L NAA + 0.5 mg/L 6-BA; T2: 2.0 mg/L NAA + 1.0 mg/L 6-BA; T3: 4.0 mg/L NAA + 2.0 mg/L 6-BA; T4: 8.0 mg/L NAA + 4.0 mg/L 6-BA. Each value represents the mean (±SD) of three experiments. Statistical analysis was performed with one-way ANOVA, and Student's *t*-tests were performed to compare the differences in mean oleanolic acid content among different treatments. *P* < 0.05 was considered statistically significant. Different letters indicate significant differences between groups.

reliability, cDNA libraries were prepared for two biological repeats of each sequencing sample. Combined with clean reads generated by our previous study [16], about 164 million clean read pairs (16.4 Gbp) were used for *de novo* transcriptome assembly. A total of 102,126 unigenes were assembled. More than 80% of clean reads could be mapped back to these assembled unigenes, indicating that the assemblies were good quality and suitable for downstream analysis (Table 1).

## Changes in gene expression profiles in leaves treated with NAA and 6-BA

Among *A. bidentata* unigenes, 53.8% and 59.4%, respectively, expressed in the Control and T samples, had RPKM values in the range of 1–100, while 44.9% and 39.3%, respectively, had RPKM values in the range of 0–1 (Fig 3). Differentially expressed genes (DEGs) between the Control and T samples were identified. Compared with Control, a total of 20,896 genes were significantly differentially expressed in the T group, among which 13,071 (62.55%) were up-

**Table 1. RNA-seq reads for four RNA-seq libraries.**

| Summary | Control-rep1 | Control-rep2 | T-rep1 | T-rep2 |
|---|---|---|---|---|
| No. of raw reads | 35,359,331 | 27,630,495 | 32,028,032 | 33,307,916 |
| No. of clean reads | 29,637,493 | 22,759,674 | 26,819,403 | 27,707,163 |
| Total % of clean reads | 83.82 | 82.37 | 83.74 | 83.18 |
| No. of mapped reads | 23,895,374 | 18,503,286 | 21,457,401 | 22,285,936 |
| Total % of mapped reads | 80.63 | 81.30 | 80.01 | 80.43 |

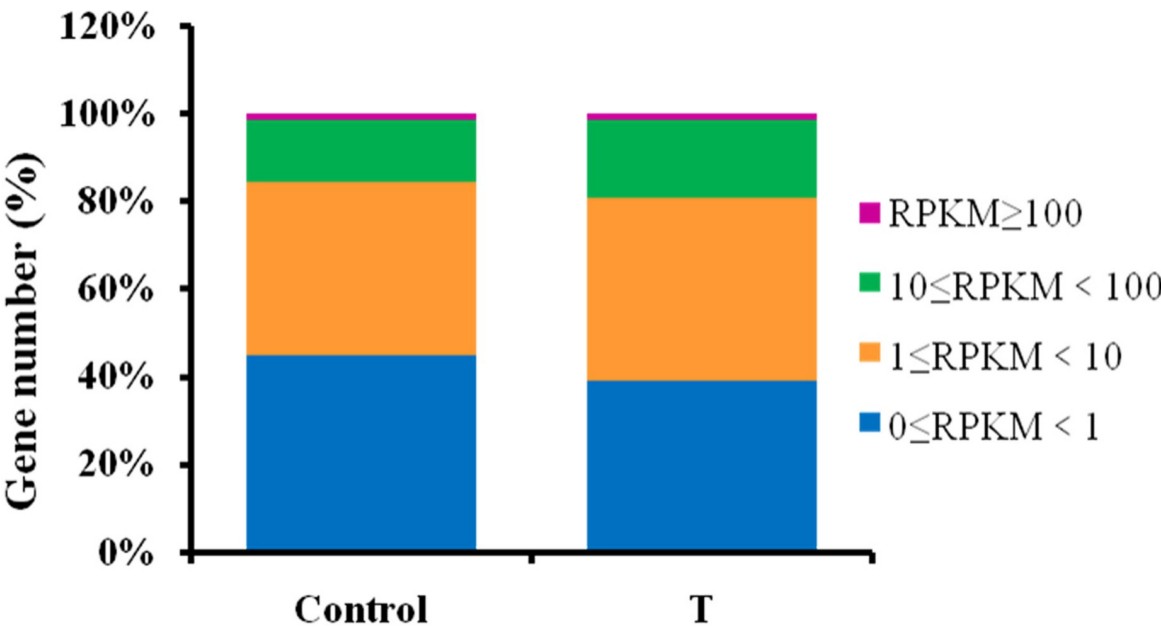

**Fig 3. Percentage of genes expressed in control and 2.0 mg/L NAA + 1.0 mg/L 6-BA-treated (T) leaves.** The different colors indicate the percentage of genes with the range of RPKM values shown in the legend.

regulated and 7,825 (37.45%) were down-regulated. These results are consistent with the higher number of unigenes in the T group than the Control group with RPKM values in the range of 1–100. These results suggest that NAA and 6-BA induced the expression of a large number of genes.

To investigate the functions of all 20,896 DEGs, GO and pathway enrichment analysis was applied. DEGs annotated in GO were grouped into 47 groups based on GO level2 classification (S1 Fig). The assigned GO terms belonged to three main ontologies: biological process, cell component, and molecular function with 22, 14, and 11 groups, respectively. In the biological process group, the most common categories were "cellular process" (11,362, 54.4%), "single-organism process" (10,434, 49.9%) and "metabolic process" (9,793, 46.9%). In the cell component group, "cell", "cell part" and "organelle" were the top three categories. Most DEGs categorized in molecular function were involved in "binding" and "catalytic activity".

To further identify metabolic or signal transduction pathways in which the DEGs are likely to be involved in promoting accumulation of oleanolic acid by JA regulation by NAA and 6-BA treatment in *A. bidentata*, pathway enrichment analysis was performed using KEGG database. For the pathway enrichment analysis, a total of 28 pathways were enriched (adjusted $P < 0.05$) with DEGs (Table 2). These pathways were divided into five categories, including metabolite degradation, photosynthesis, carbohydrate metabolism, terpenoid biosynthesis, and biosynthesis of other compounds. In the metabolite degradation pathways, 58 DEGs were predicted to be involved in processes that detoxify hydrogen peroxide, including baicalein, betanidin, and luteolin triglucuronide degradation processes. For instance, the expression levels of 43 DEGs encoding peroxidase were significantly increased in the T group compared with Control. Photosynthesis metabolic pathways, such as oxygenic photosynthesis, NAD-ME-type and PEPCK-type photosynthetic carbon assimilation cycle, the Calvin-Benson-Bassham cycle, and the chlorophyll cycle were enriched in DEGs. In treated leaves, most DEGs involved in oxygenic photosynthesis, the Calvin-Benson-Bassham cycle and the chlorophyll cycle were down-regulated except for 3 transketolases. However, 12 malic enzyme genes,

**Table 2. Pathway enrichment analysis of genes differentially expressed in phytohormone-treated *A. bidentata* leaves.**

| Pathway ID | Pathway Name | Nd | Bn | p value | adj P |
|---|---|---|---|---|---|
| **Degradation** | | | | | |
| PWY-7214 | baicalein degradation (hydrogen peroxide detoxification) | 58 | 106 | 2.24E-06 | 1.51E-04 |
| PWY-5461 | betanidin degradation | 58 | 106 | 2.24E-06 | 1.51E-04 |
| PWY-7445 | Luteolin triglucuronide degradation | 58 | 106 | 2.24E-06 | 1.51E-04 |
| PWY-1081 | homogalacturonan degradation | 30 | 49 | 4.03E-05 | 1.26E-03 |
| MANNCAT-PWY | D-mannose degradation | 5 | 5 | 3.86E-03 | 4.82E-02 |
| **Photosynthesis** | | | | | |
| PHOTOALL-PWY | oxygenic photosynthesis | 38 | 64 | 1.06E-05 | 5.96E-04 |
| PWY-7115 | photosynthetic carbon assimilation cycle, NAD-ME type | 33 | 60 | 3.15E-04 | 6.64E-03 |
| PWY-7117 | photosynthetic carbon assimilation cycle, PEPCK type | 30 | 52 | 1.89E-04 | 4.26E-03 |
| CALVIN-PWY | Calvin-Benson-Bassham cycle | 28 | 49 | 3.86E-04 | 7.64E-03 |
| PWY-5068 | chlorophyll cycle | 9 | 9 | 4.48E-05 | 1.26E-03 |
| **Carbohydrate metabolism** | | | | | |
| GLUCONEO-PWY | gluconeogenesis I | 39 | 80 | 2.17E-03 | 3.18E-02 |
| PWY-4821 | UDP-D-xylose biosynthesis | 17 | 18 | 7.28E-08 | 1.23E-05 |
| PWY-7346 | UDP-alpha-glucuronate biosynthesis (from UDP-glucose) | 5 | 5 | 3.86E-03 | 4.82E-02 |
| **Terpenoid biosynthesis** | | | | | |
| PWY-5910 | superpathway of geranylgeranyl diphosphate biosynthesis I (via mevalonate) | 36 | 61 | 2.19E-05 | 9.23E-04 |
| PWY2OL-4 | superpathway of linalool biosynthesis | 29 | 45 | 1.32E-05 | 6.34E-04 |
| PWY-7182 | linalool biosynthesis I | 28 | 44 | 2.64E-05 | 9.87E-04 |
| PWY-7141 | linalool biosynthesis II | 21 | 37 | 2.32E-03 | 3.25E-02 |
| PWY-5122 | geranyl diphosphate biosynthesis | 20 | 36 | 4.11E-03 | 4.95E-02 |
| PWY-6475 | *trans*-lycopene biosynthesis II (plants) | 19 | 33 | 2.98E-03 | 4.02E-02 |
| PWY-922 | mevalonate pathway I | 15 | 22 | 7.36E-04 | 1.31E-02 |
| PWY-7186 | superpathway of scopolin and esculin biosynthesis | 10 | 13 | 1.46E-03 | 2.34E-02 |
| PWY-5725 | farnesene biosynthesis | 8 | 8 | 1.36E-04 | 3.29E-03 |
| PWY-6275 | beta-caryophyllene biosynthesis | 8 | 8 | 1.36E-04 | 3.29E-03 |
| PWY-5670 | epoxysqualene biosynthesis | 6 | 6 | 1.27E-03 | 2.14E-02 |
| **Others** | | | | | |
| PWY0-541 | cyclopropane fatty acid (CFA) biosynthesis | 26 | 27 | 4.42E-12 | 1.49E-09 |
| PWY-6163 | chorismate biosynthesis from 3-dehydroquinate | 17 | 26 | 6.79E-04 | 1.27E-02 |
| PWY-7498 | phenylpropanoids methylation (ice plant) | 13 | 19 | 1.61E-03 | 2.47E-02 |
| PWY-2181 | free phenylpropanoid acid biosynthesis | 9 | 9 | 4.48E-05 | 1.26E-03 |

Bn (Background number) indicates the total number of unigenes present in each pathway. Nd (Number of DEGs) indicates the number of differentially expressed unigenes in each pathway.

6 out of 10 aspartate aminotransferase genes (*AST*), and 8 phosphoenolpyruvate carboxykinase genes (*PEPCK*) were up-regulated in the T group, and are involved in NAD-ME-type and PEPCK-type photosynthetic carbon assimilation cycle metabolic pathways. These results and the reduction in the levels of the photosynthetic pigments chlorophyll *a* and *b* and carotenoid (S2 Table) suggest that photosynthesis was inhibited by hormone treatment.

For carbohydrate metabolism, there were three categories enriched in DEGs: gluconeogenesis I, UDP-D-xylose biosynthesis, and UDP-alpha-D-glucuronate biosynthesis. Genes encoding malic enzyme and phosphoenolpyruvate carboxylase, which participate in the gluconeogenesis I pathway, were up-regulated in treated leaves. All DEGs involved in UDP-D-xylose biosynthesis and UDP-alpha-D-glucuronate biosynthesis were up-regulated. For terpenoid metabolism,

the top three pathways enriched in DEGs were the geranylgeranyl diphosphate biosynthesis I super pathway, the linalool biosynthesis super pathway, and linalool biosynthesis I. It is worth noting that mevalonate pathway I, farnesene biosynthesis, and epoxysqualene biosynthesis pathway were enriched in DEGs, and the expression levels of 26 out of 29 DEGs in these pathways were significantly increased in treated leaves. For instance, the expression level of *HMGR*, which encodes the first rate-limiting enzyme involved in the MVA pathway, was 501.66 times higher in T than in Control leaves. These results indicate that triterpenoid biosynthesis is induced by NAA and 6-BA treatment. For biosynthesis of other compounds, the cyclopropane fatty acid, chorismate, and free phenylpropanoid acid biosynthesis pathways were enriched in DEGs. All DEGs involved in these pathways were down-regulated except for the cyclopropane fatty acid synthase gene. The results of pathway enrichment analysis suggest that the expression of *A. bidentata* genes involved in photosynthesis metabolic pathways and terpenoid metabolic processes are affected by the combination of 2.0 mg/L NAA and 1.0 mg/L 6-BA. In addition, this is the first report of C4 type photosynthetic carbon assimilation cycle in *A. bidentata* suggesting that *A. bidentata* may be a C4 plant. About 28% of the 900 species in the Amaranthaceae estimated to occur in the family are C4 species, such as *Gomphrena meyeniana*, *Amaranthus hypochondriacus* [45].

### DEGs involved in JA signaling pathways

Seventeen candidate DEGs were predicted to be involved in JA signaling pathways (Table 3), which regulate the synthesis of secondary metabolites [46]. Most DEGs involved in JA signaling were up-regulated. Our results suggest that the expression of JA-responsive genes was affected by treatment with 2.0 mg/L NAA + 1.0 mg/L 6-BA.

### Transcription factors differentially expressed in response to auxin and cytokinin treatment

Several TF families such as MYC, MYB, WRKY, and AP2/ERF are involved in regulating secondary metabolism in different medicinal plants [28]. Therefore, we asked whether DEGs included TFs. We found that 308 TFs were differently expressed in the T group compared with the Control group, including 17 MYBs, 21 bHLHs, 14 WRKYs, and 21 bZIPs (Table 4 and S3 Table). Many more TFs were up-regulated than down-regulated. WRKYs and bZIPs were mainly down-regulated, while AP2-ERFs, ERFs, bHLH, MADS, heat stress transcription factors, GATAs, and Nuclear transcription factor Y were mainly up-regulated.

### DEGs involved in the oleanolic acid biosynthesis pathway

In the present study we identified a total of 395 unigenes (S4 Table) encoding almost all the enzymes known to be involved in oleanolic acid biosynthesis via the MVA and MEP pathways, including 200 DEGs. The DEGs related to oleanolic acid metabolism are schematically

**Table 3. Differentially expressed genes in response to NAA and 6-BA treatment in *A. bidentata*.**

| JA Signaling | Number of DEGs | Up-regulated genes | Down-regulated genes |
|---|---|---|---|
| Jasmonate ZIM domain protein | 7 | 7 | 0 |
| Jasmonate-induced protein | 2 | 2 | 0 |
| Methyl esterase 1 | 4 | 1 | 3 |
| Ninja-family protein | 2 | 2 | 0 |
| Topless-related protein | 2 | 2 | 0 |
| Total | 17 | 14 | 3 |

**Table 4. Transcription factors differentially expressed in response to auxin and cytokinin treatment in *A. bidentata*.**

| TF family | Number of DEGs | Up-regulated TF genes | Down-regulated TF genes |
|---|---|---|---|
| MADS box transcription factor | 34 | 27 | 7 |
| Ethylene-responsive transcription factor | 31 | 20 | 11 |
| AP2-like ethylene-responsive transcription factor | 21 | 19 | 2 |
| bHLH transcription factor | 21 | 16 | 5 |
| bZIP transcription factor | 21 | 7 | 14 |
| Heat stress transcription factor | 19 | 12 | 7 |
| Nuclear transcription factor Y | 17 | 16 | 1 |
| MYB transcription factor | 17 | 10 | 7 |
| WRKY transcription factor | 14 | 4 | 10 |
| GATA transcription factor | 7 | 5 | 2 |
| Others | 106 | 77 | 29 |
| Total | 308 | 213 | 95 |

represented in Fig 4. In the MVA pathway, the transcription level of unigenes involved in terpenoid precursor biosynthesis, such as *HMGS* (HMG-CoA synthase), *HMGR* (HMG-CoA reductase), *PMK* (phosphomevalonate kinase), *MDD* (mevalonate-5-diphosphate decarboxylase), and *IDI* (isopentenyl diphosphate isomerase), were up-regulated by 2.0 mg/L NAA and 1.0 mg/L 6-BA. In contrast, most terpenoid precursor synthesis genes in the MEP pathway, such as *DXS* (1-deoxy-D-xylulose5-phosphate synthase), *CMS* (2-C-methyl-D-erythritol 4-phosphate cytidylyl-transferase), *HDS* (4-hydroxy-3-methylbut-2-en-1-yl diphosphate synthase), and *HDR* (4-hydroxy-3-methylbut-2-enyl diphosphate reductase), were down-regulated by NAA and 6-BA treatment. Triterpenoid skeleton synthesis genes, such as *FPS*, *SS*, *SE* and *β-AS* were also up-regulated in the T group. In the stage when triterpenoid skeletons are modified, the hydroxylation and glycosylation processes, which are catalyzed by cytochrome P450 monooxygenases and glycosyl transferases in turn, are important for the production of triterpenoid saponins. In this study, 186 unigenes were annotated as CYP450s, including 41 up-regulated and 48 down-regulated unigenes in the T group. In order to further narrow the potential range of CYP450s involved in biosynthesis of oleanolic acid saponin, 22 up-regulated CYP450s (with amino acid sequence lengths > 450) were selected for phylogenetic analysis from *A. bidentata* treated leaves. As a template, 245 CYP450 annotated genes in the *Arabidopsis* genome were obtained from http://www.p450.kvl.dk/p450.shtml. Twenty-two unigenes (S6 Table) from *A. bidentata* were clustered in the CYP71 (10 unigenes), CYP711 (5 unigenes), CYP85 (3 unigenes), CYP72 (2 unigenes), and CYP86 (2 unigenes) clans (S2 Fig). After phylogenetic analysis of five unigenes and fifty-five CYP450s (S5 Table) related to triterpene synthesis (S4 Fig), and the CYP450s with similar functions were further analyzed with five unigenes. Among them, unigene UN082587 was annotated to *P. ginseng* CYP716A52v2 (Fig 5). This result indicated that the UN082587 gene might encode an enzyme that converts β-amyrin to oleanolic acid. Surprisingly, UN082587 was 117.8 times higher in T than Control leaves. Three unigenes (UN046523, UN046524 and UN085763) were homologous to *Glycyrrhiza uralensis* CYP88D6 (β-amyrin 11-oxidase). A total of 77 DEGs encoding glycosyl transferases were identified, 35 of which were up-regulated in treated leaves. These results indicate that the expression level of most genes involved in oleanolic acid biosynthesis were up-regulated by NAA and 6-BA treatment.

## RNA-seq accuracy determined by real-time PCR

To confirm the reliability of the RNA-seq data, the expression levels of six DEGs involved in the oleanolic acid biosynthesis pathways (*HMGR*, *PMK*, *FPS*, *SS*, *SE*, and *β-AS*) were validated

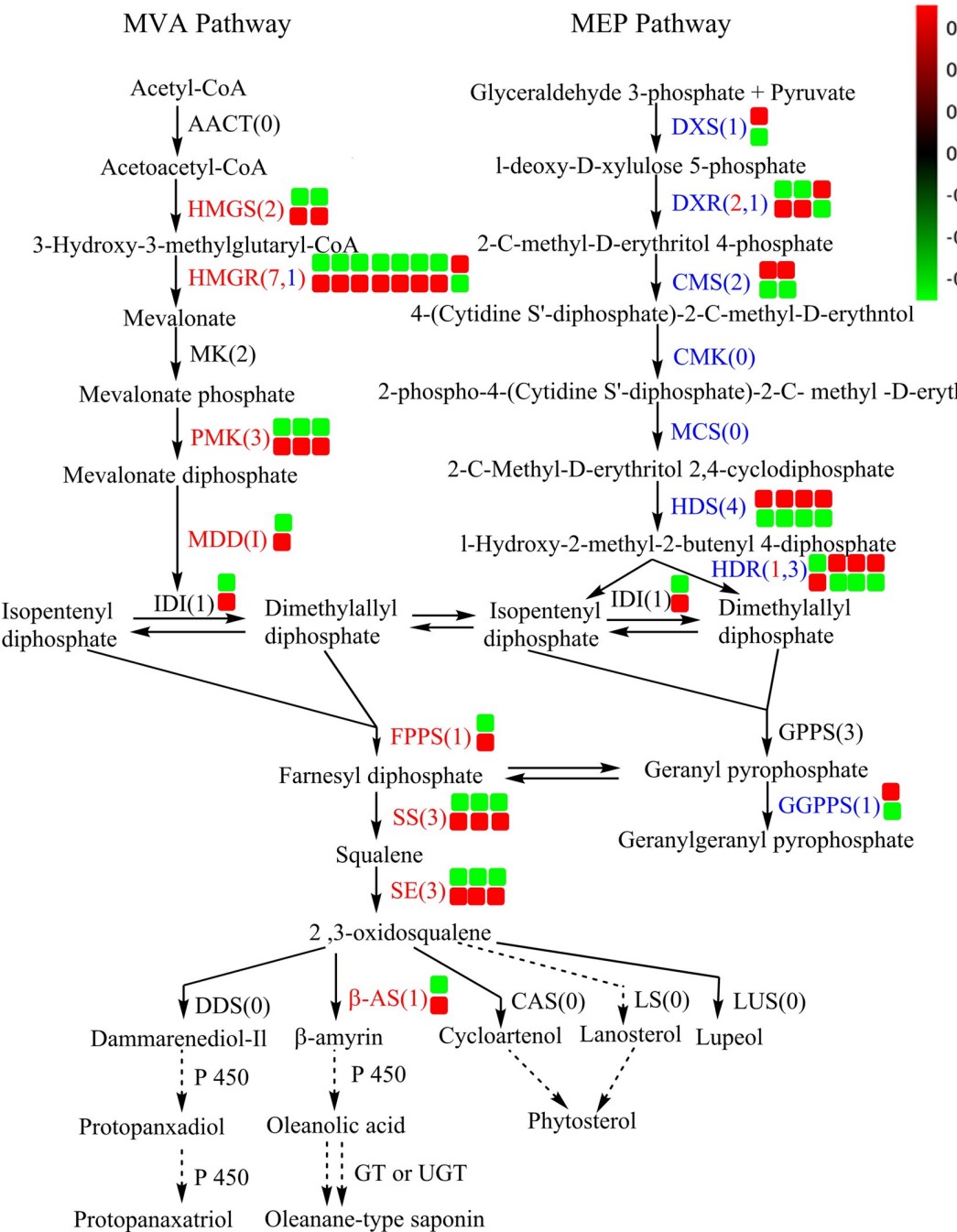

## Figure 4

**Fig 4. Overview of the triterpenoid biosynthetic pathway.** For each gene, the top row of squares represents the Control and the bottom row represents the NAA and 6-BA treatment. The number of squares indicates the number of differentially expressed genes with fold changes ≥ 2 or fold change ≤ 0.5 at adjusted $P < 0.05$. Different colors indicate genes that were down-regulated (green) or up-regulated (red) in the T group compared with the Control group. Abbreviations are as follows: AACT, acetyl-CoA acetyltransferase; MK, mevalonate kinase; DXR, 1-deoxy-D-xylulose5-phosphate reductoisomerase; CMK, 4-(cytidine-diphospho)-2-C-methyl-D-erythritol kinase; MCS, 2-C-methyl-D-erythritol-2,4- cyclodiphosphate synthase; GPPS, geranyl diphosphate synthase; GGPPS, geranylgeranyl diphosphate synthase; DDS, dammarenediol synthase; LS, lanosterol synthase; LUS, lupeol synthesis; and UGT, UDP-glycosyltransferase.

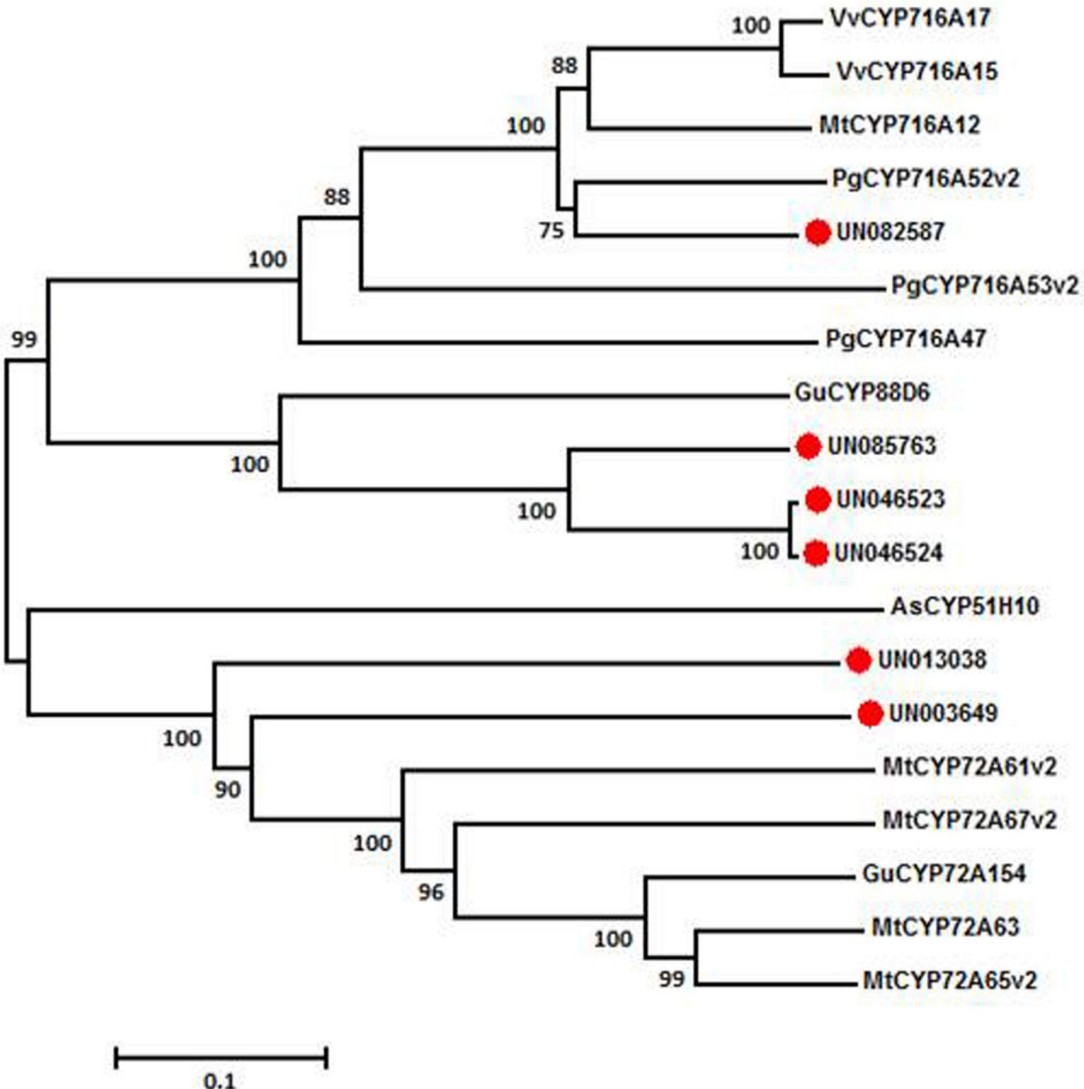

**Fig 5. Phylogenetic tree of the *A. bidentata* CYP450s.** The phylogenetic tree was constructed based on the deduced amino acid sequences for the *A. bidentata* CYP450s (red circle) and other plant CYP450s involved in triterpenoid biosynthesis. The protein sequence ID was retrieved from the NCBI GenBank using the accession numbers marked in red provided in S5 Table.

using qRT-PCR. The expression patterns of these genes in the T and Control leaves were consistent with the expression patterns based on RNA-seq, and there was a close correlation (r = 0.94) between the expression changes determined by RNA-seq and by qRT-PCR (Fig 6).

## Discussion

### JA-mediated transcriptional regulation of oleanolic acid metabolism

In the present study, the levels of JA, stress hormones that regulate the synthesis of secondary metabolites, increased in *A. bidentata* leaves treated with 2.0 mg/L NAA + 1.0 mg/L 6-BA (S3 Fig). This is consistent with a previous finding that endogenous JA concentration was significantly increased by 2.65-fold after NAA treatment in *Chlorella vulgaris* [46]. The increase in

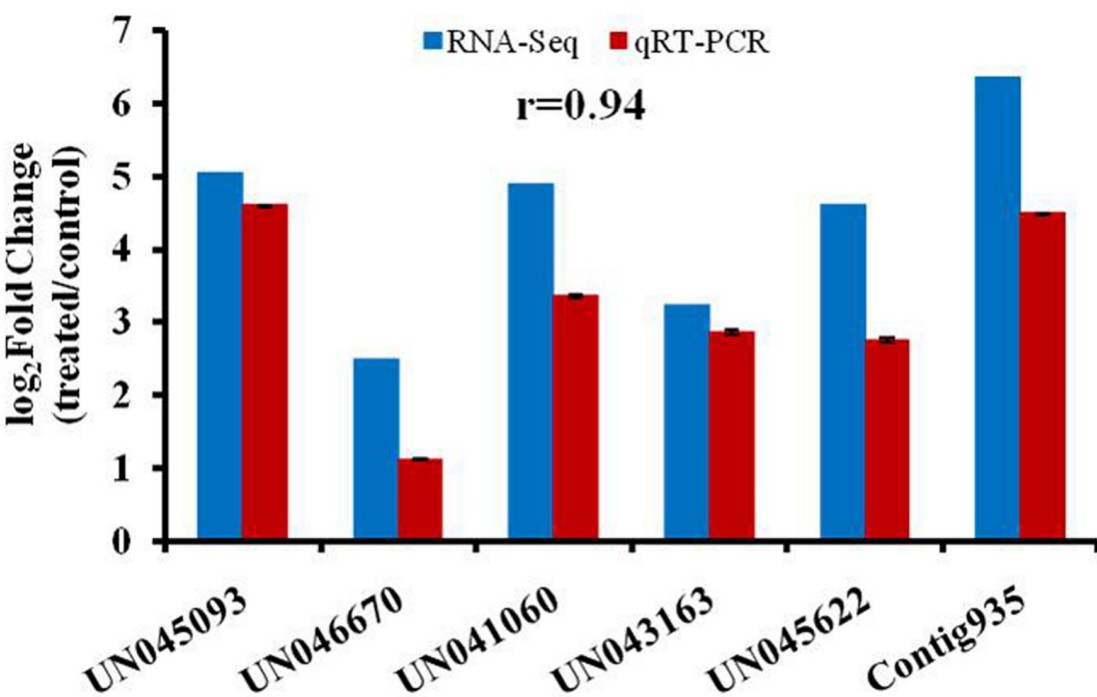

**Fig 6. Expression patterns of genes involved in the triterpenoid saponin and phytosterols biosynthesis pathways determined by RNA-seq and qRT-PCR.**

JA may be due to direct activation of the JA signaling pathway or may be indirectly caused by the accumulation of IAA due to the collaborative relationship between IAA and JA [47].

JA ZIM domain (JAZ) is a key molecule that serves as the on/off switch for JA signaling. In the presence of JA or its bioactive derivatives, JAZ proteins are degraded, freeing TFs for expression of specific sets of JA-respective genes, thereby promoting physiological activity, including producing specific sets of secondary metabolites [48]. In addition to the F-box protein COI1, some transcription factors were JAZ interactors including bHLH TFs, R2R3 MYB TFs, and TFs of other hormone signaling pathways [49]. Multiple experiments have demonstrated that MYC, MYB, WRKY and AP2/ERF TFs are involved in the regulation of terpenoid metabolism in different medicinal plants [28, 50]. For example, AaWRKY1, TSAR1 (triterpene saponin activating regulator), and TSAR2 regulate *HMGR* expression involved in the MVA pathway as a rate-limiting enzyme in *Artemisia annua* and *Medicago truncatula* [50, 51]. The bHLH transcription factors TSAR1 and TSAR2 are two homologous jasmonate-inducible transcription factors [51]. In *M. truncatula* hairy roots, overexpression of *TSAR1* and *TSAR2* resulted in significantly elevated transcript levels of the MVA pathway genes and all consecutive genes involved in the generation of the β-amyrin backbone, such as *HMGS*, *HMGR1*, *SS*, and *β-AS* [51, 52]. GbWRKY1 regulates *GbMVD* expression involved in terpene trilactone biosynthesis by binding to the W-box in *Ginkgo biloba* [53]. In the present study, most AP2/ERF TFs, bHLH TFs, and MYB TFs were up-regulated in the T group, with significantly increased transcript levels of terpenoid biosynthesis genes, especially *HMGR*, *SS*, *SE*, and *β*-AS in *A. bidentata* (Fig 7). Therefore, JA regulates oleanolic acid accumulation by freeing TFs and increasing expression of terpenoid biosynthesis genes in *A. bidentata*. These results were consistent with overexpression of *TSARs* to promote oleanane-type triterpene saponin in *M. truncatula*.

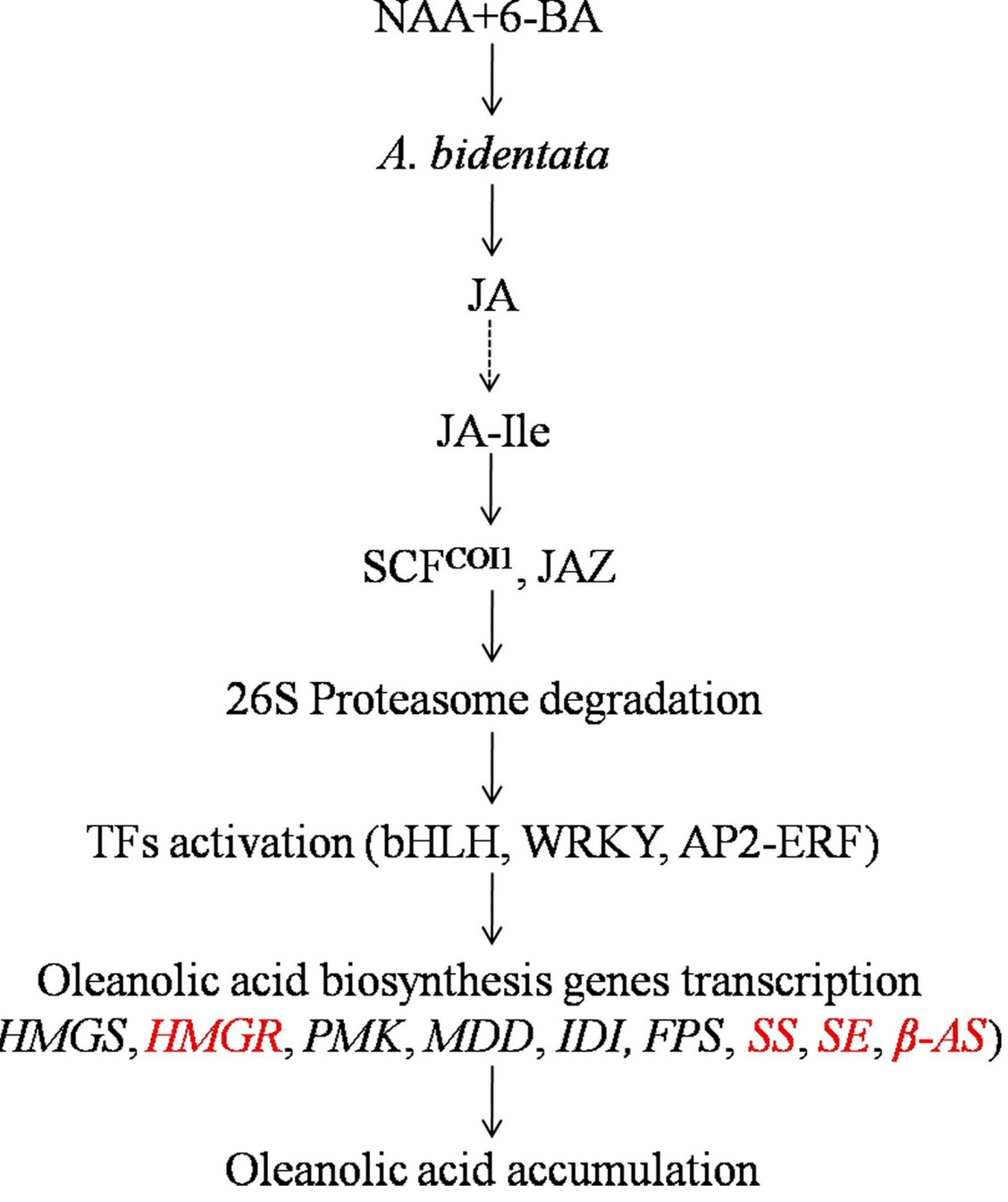

**Fig 7. NAA and 6-BA promote accumulation of oleanolic acid by JA regulation in *A. bidentata*.**

## NAA and 6-BA promote terpenoid biosynthesis and photosynthesis processes

For the pathway enrichment analysis of DEGs, most metabolic pathways were classified as terpenoid biosynthesis or photosynthesis. For terpenoid biosynthesis, three linalool biosynthesis pathways and geranyl diphosphate biosynthesis pathway were identified, and most DEGs involved in their biosynthesis pathway were down-regulated, especially 1, 4-dihydroxy-

2-naphthoate (DHNA) polyprenyltransferase. Previous studies demonstrated that DHNA polyprenyltransferase belongs to the prenyltransferases family of aromatic prenyltransferases and catalyzes the transfer of prenyl moieties to aromatic acceptor molecules, forming C–C bonds between C-1 or C-3 of the isoprenoid substrate and one of the aromatic carbons of the acceptor substrate [54, 55]. The prenyl moiety is derived from allylic isoprenyl diphosphates including dimethylallyl diphosphate (DMAPP; C5), geranyl diphosphate (GPP; C10), and farnesyl diphosphate (FPP; C15). These allylic isoprenyl diphosphates were substrates of triterpenoid biosynthesis. Therefore, more and more allylic isoprenyl diphosphates were involved in triterpenoid biosynthesis and promoted accumulation of oleanolic acid due to low expression levels of 1, 4-dihydroxy-2-naphthoate polyprenyltransferase in the T group. For photosynthesis, most genes of malic enzyme, *AST* and *PEPCK* were up-regulated and involved in NAD-ME-type and PEPCK-type photosynthetic carbon assimilation cycle metabolic pathways in treated leaves. The use of C4 pathway is conducive to the use of low concentrations of carbon dioxide, enhancing plant stress resistance. Therefore, hormones affect the level of endogenous JA and promote oleanolic acid biosynthesis.

## Conclusions

Transcriptomes of *A. bidentata* leaves sprayed with 2.0 mg/L NAA + 1.0 mg/L 6-BA allowed us to gain insight into the molecular mechanisms of the response to phytohormone treatment. A total of 20,896 DEGs were identified, and 62.55% of these DEGs were up-regulated by phytohormone treatment. The pathway enrichment analysis demonstrated that photosynthesis metabolic pathways and terpenoid biosynthesis were overrepresented by genes differentially expressed in the treatment group, suggesting that photosynthesis and terpenoid metabolic processes are affected by treatment with 2.0 mg/L NAA and 1.0 mg/L 6-BA. Furthermore, the expression levels of genes involved in JA signaling were differentially expressed between Control and T. The endogenous JA content of treated leaves also significantly increased over time, suggesting that endogenous JA levels were regulated by exogenous phytohormones. In conclusion, exogenous phytohormones work in conjunction with endogenous phytohormones to promote the accumulation of oleanolic acid in *A. bidentata*.

## Supporting information

**S1 Fig. GO enrichment analysis for the differentially expressed genes in *A. bidentata*.**
(TIF)

**S2 Fig. Phylogenetic tree constructed based on the deduced amino acid sequences of the *A. bidentata* CYP450s and all *Arabidopsis* CYP450s.** The phylogenetic tree was generated using the neighbor-joining (NJ) method in MEGA6.
(TIF)

**S3 Fig. Effects of exogenous NAA and 6-BA on JA contents in *A. bidentata* leaves.**
(TIF)

**S4 Fig. Phylogenetic tree constructed based on the deduced amino acid sequences of the five *A. bidentata* CYP450s and 55 CYP450s related to triterpene synthesis.** The phylogenetic tree was generated using the neighbor-joining (NJ) method in MEGA6.
(TIF)

**S1 Table. Primers used for qRT-PCR analysis.**
(DOCX)

**S2 Table. Photosynthetic pigment content in leaves.**
(DOCX)

**S3 Table. List of transcription factors.**
(DOCX)

**S4 Table. List of candidate genes encoding enzymes in the oleanolic acid biosynthesis pathways.**
(DOCX)

**S5 Table. List of fifty-five CYP450s related to triterpene synthesis in constructing phylogenetic tree.** The CYP450s marked in red represent those used in the construction of Fig 5.
(DOCX)

**S6 Table. List of twenty-two unigenes of *A. bidentata* used to build phylogenetic tree.** The unigenes marked in red represent those used in the construction of Fig 5.
(DOCX)

## Author Contributions

**Conceptualization:** Yanqing Liu, Li Tang, Can Wang, Jinting Li.

**Formal analysis:** Yanqing Liu, Can Wang.

**Investigation:** Li Tang, Can Wang.

**Resources:** Jinting Li.

**Writing – original draft:** Yanqing Liu.

**Writing – review & editing:** Li Tang, Jinting Li.

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
