## [Decision Letter · Decision Letter 0]

4 Sep 2019

PONE-D-19-21410

NAA and 6-BA promote accumulation of oleanolic acid by JA regulation in Achyranthes bidentata Bl. through de novo transcriptomics

PLOS ONE

Dear Dr Li, 

Thank you for submitting your manuscript to PLOS ONE. After careful consideration by two experts in the field, we feel that it has merit but does not fully meet PLOS ONE’s publication criteria as it currently stands. Therefore, we invite you to submit a revised version of the manuscript that addresses the various points raised by the two reviewers. 

We would appreciate receiving your revised manuscript by the end of september. To enhance the reproducibility of your results, we recommend that if applicable you deposit your laboratory protocols in protocols.io, where a protocol can be assigned its own identifier (DOI) such that it can be cited independently in the future. For instructions see: http://journals.plos.org/plosone/s/submission-guidelines#loc-laboratory-protocols

We look forward to receiving your revised manuscript.

Kind regards,

Marie-Joelle Virolle, PhD

Academic Editor

PLOS ONE

Journal Requirements:

3. In your Methods section, please provide additional details regarding the A. bidentata seeds used in your study and ensure you have described the source. For more information regarding PLOS' policy on materials sharing and reporting, see https://journals.plos.org/plosone/s/materials-and-software-sharing#loc-sharing-materials

4. We note that you are reporting an analysis of a microarray, next-generation sequencing, or deep sequencing data set. PLOS requires that authors comply with field-specific standards for preparation, recording, and deposition of data in repositories appropriate to their field. Please upload these data to a stable, public repository (such as ArrayExpress, Gene Expression Omnibus (GEO), DNA Data Bank of Japan (DDBJ), NCBI GenBank, NCBI Sequence Read Archive, or EMBL Nucleotide Sequence Database (ENA)). In your revised cover letter, please provide the relevant accession numbers that may be used to access these data. For a full list of recommended repositories, see http://journals.plos.org/plosone/s/data-availability#loc-omics or http://journals.plos.org/plosone/s/data-availability#loc-sequencing.

Reviewers' comments:

Reviewer's Responses to Questions

**Comments to the Author**

1. Is the manuscript technically sound, and do the data support the conclusions?

Reviewer #1: Partly

Reviewer #2: Partly

2. Has the statistical analysis been performed appropriately and rigorously? 

Reviewer #1: Yes

Reviewer #2: Yes

3. Have the authors made all data underlying the findings in their manuscript fully available?

Reviewer #1: Yes

Reviewer #2: Yes

4. Is the manuscript presented in an intelligible fashion and written in standard English?

Reviewer #1: Yes

Reviewer #2: No

5. Review Comments to the Author

Reviewer #1: The manuscript by Liu and co-workers describes the investigation of the oleanolic acid biosynthesis in the Chinese medicinal plant Achyranthes bidentata. Compared to control plants, A. bidentata plants exogenously treated with NAA and 6BA accumulated higher amounts of oleanolic acid, a major phytochemical in the roots of the plant. RNA-Seq analysis of control and treated plants revealed that almost all genes encoding enzymes involved in oleanolic acid biosynthesis were upregulated upon NAA and 6BA treatment. Furthermore, also genes involved in JA signaling and a large amount of transcription factors were NAA/6BA-responsive. As also an increased JA accumulation was observed, the authors conclude that combined auxin and cytokinin treatment leads to increased oleanolic acid accumulation via a JA-mediated signaling cascade. To my opinion this conclusion is not supported by the provided evidence (for more details, see my specific comments below).

Remarks:

1/ Lines 47-50 needs a reference.

2/ Lines 52-54: OSCs are not generally known as rate-limiting enzymes in the biosynthesis of sterols and/or triterpenoids. Rather, HMGR and SQE are. To be either corrected or supported by references that support the notion of OSCs being rate-limiting enzymes.

3/ Line 56: Cholesterol? Apart from Solanaceae, cholesterol is only a minor plant sterol.

4/ Lines 87-89: explain better the JA signaling mechanism; it is a bit too summarily described here.

5/ Lines 104-107: how exactly were the plants treated? Spraying? If so, how much (volume) was sprayed on the plants?

6/ Lines 126-128: this sentence is not clear to me.

7/ Lines 252-254: this is not clear to me, how can you have such high percentages for the different processes, unless the genes are part of different processes?

8/ Lines 297-299: nice observation, it could be supported by the fact that other amaranthaceae are known to be C4 plants.

9/ Lines 313-314: Use bZIP and bHLH as abbreviation.

10/ Lines 346-347: What is the reason the CYP88D6 orthologs were pointed out? Likely the encountered CYP88 genes are involved in ent-kaurene biosynthesis rather than triterpene biosynthesis. Furthermore, oxidation at the C-11 position is not required for oleanolic acid biosynthesis.

11/ Lines 347-349: What is the reason the differentially expressed UGTs are indicated? No UGTs are required for the oleanolic acid biosynthesis.

12/ Remark Figure 4. I don’t entirely agree with the downstream MVA pathway presented in Figure 4. There is a differentiation between the cytosolic FPP production, which is done by a single FPS enzyme that condenses two units of IPP and one unit of DMAPP and the plastidial FPP production, which is catalyzed by two distinct enzymes, GPPS and FPS. The cytosolic pathway leads to sesquiterpenes and triterpenes, the plastidial pathway leads to mono-, di- and tetraterpenes. In this respect it seems to me that all plastid-derived metabolite biosynthesis pathways (linalool = monoterpene) are down, whereas all cytosolic (triterpene and farnesene = sesquiterpene) are upregulated by the NAA/6BA treatment. Another remark on figure 4 is the inclusion of the ginsenoside biosynthesis. Why is this included here? Are ginsenosides reported in A. bidentata?

13/ Remark Figure 5: Why were these specific P450s chosen? Many more have been shown to be involved in triterpene biosynthesis. Either provide an explanation for the limited choice of P450s or be more complete and include all P450s involved in triterpene or oleanane biosynthesis.

14/ Line 374: A link is made between JAZ repressors and transcription factors (like bHLH) that bind these JAZ proteins. However, the TSAR transcription factors, for instance, do not have a JAZ interaction domain (JID) and as such they don’t interact with JAZ. Yet they are involved in the regulation of triterpene biosynthesis.

15/ Lines 389-391: The conclusion that JA regulates oleanolic acid accumulation is not supported by the provided evidence. You show that JA levels and JA response are upregulated upon NAA/6BA treatment and that triterpene levels are upregulated upon NAA/6BA treatment. This is only a correlation, not necessarily a causation. To point to a possible involvement of JA signaling, it would be more convincing to carry out a time-dependent transcript profiling of some JAZ/TF/JA genes, and some triterpene biosynthesis genes. Only if the JAZ/TF/JA genes are upregulated before the triterpene biosynthesis genes you can point to a possible role of the JA signaling in the increased biosynthesis of oleanolic acid. Another option is to carry out a JA-treatment of your plants and measure the effect of it on oleanolic acid accumulation or transcription of OA biosynthesis genes. In a perfect world you could also work in JA biosynthesis or signaling mutants, but this is clearly impossible in your plant species…

16/ Lines 410-411: tertripenoid?

Reviewer #2: The research subject of this manuscript is interesting and valuable because data on regulation of triterpenoid biosynthesis in medicinal plants are still scarce. However, I have several doubts about the experimental model, moreover, some obtained results are difficult to understand.

Line 40. „Oleanolic acid, a pentacyclic triterpenoid saponin”. Oleanolic acid is not a saponin! It is an aglycon of numerous saponins, but it can also occur in plants in a free form. The Authors are not explaing if Achyrantes bidentata contains oleanolic acid in a free form or in the form of glycosides (saponins). It is rare that there is only one saponin in the plant, usually there are several, with different sugar chains. Moreover, the Authors are not describing the details of extraction and HPLC metods which they used for oleanolic acid determination (the reference 29 does not contain a suitable description of these procedures!). It is not clear if extracts were purified, if they were hydrolyzed (to release oleanolic acid from its glycosides), what conditions were used for HPLC etc.

Line 103. „Seeds of A. bidentata were planted and grown in the natural environment at the

Wenxian Agricultural Science Institute of Henan Province, China.” Why these conditions („natural”) have been chosen? Plants were then growing during quite a long time (e.g., 20 days after treatment), they could be exposed to the influence of various abiotic and biotic stresses (drought, temperature, light, herbivore attack, pathogen infection), which can change the biosynthesis and accumulation of oleanolic acid as a result of the response of secondary plant metabolism to stress conditions. So it cannot be concluded that the changes in oleanolic acid content are only due to the hormonal treatment. It is true that Authors are comparing the treated samples with the controls, but it cannot be excluded that the controls had slightly different growth conditions (the spraying with hormones could for example act as an attractant or repelent for some insects or microorganisms). Authors should explain why they have chosen „natural” conditions for their experiments, instead of the controlled conditions in a green house.

Line 202. „After 20 days of treatment, the root length, fresh weight of root, dry weight of root, plant height, fresh weight of plant, and dry weight of plant had the same trend of first increasing then decreasing”. This is very difficult to understand and hard to believe. The plants were suddenly shrinking? How the root length and plant height can be at first bigger, and than smaller?

Line 204. „Lower concentrations of NAA and 6-BA improved these indexes while higher concentrations inhibited these indexes”. This is an awkward expression. Indexes cannot be inhibited. Phenomenon or processes can, but not indexes which are just the parameters.

Line 257. For the pathway enrichment analysis, a total of 28 pathways were enriched. What do it mean?

Line 318.” …almost all the enzymes known to be involved in oleanolic acid biosynthesis via the MVA and MEP pathways..”. The participation of MEP pathway in oleanolic acid biosynthesis – in normal conditions - is very doubful. The possibility of the cross-talk between these two pathway cannot be ruled out but it usually occurs in special circumstances, particularly when the MVA pathway is blocked or inhibited. Do the Authors expect such conditions after applied treatment?

The style of writing is sometimes a little chaotic and full of repetitions.

6. PLOS authors have the option to publish the peer review history of their article (what does this mean?). If published, this will include your full peer review and any attached files.

Reviewer #1: Yes: Jacob Pollier

Reviewer #2: No

---

## [Author Response · Author response to Decision Letter 0]

29 Sep 2019

Responds to academic editor:

1.I have carefully checked the manuscript, and the manuscript has met PLOS ONE's style requirements, including those for file naming.

2.I have modified the manuscript for language usage, spelling, and grammar and sincerely hope that it can meet the requirements of the magazine.

My colleague, Li Tang, and I participated in the revision of the manuscript,and my name is Yanqing Liu.

For the resubmission, I have provide the following files:

“Response to Reviewers”;

“Revised Manuscript with Track Changes” (uploaded as a *supporting information* file) and a clean copy of the edited manuscript (uploaded as the new *manuscript* file).

3.In my Methods section, I have provided additional details regarding the A. bidentata seeds used in the study.

4.In the revised cover letter, I have provided the relevant accession numbers that may be used to access data. All the readings obtained in this study have been uploaded to the NCBI Sequence Read Archive, and the accession number is PRJNA350183.

Responds to reviewers:

Reviewer #1

1/ Lines 47-50 

I have added the following references:

Miettinen K, Iñigo S, Kreft L, Pollier J, De Bo C, Botzki A, et al. The TriForC database: a comprehensive up-to-date resource of plant triterpene biosynthesis. Nucleic Acids Res. 2018; 46(D1): D586-D594. https://doi.org/10.1093/nar/gkx925 PMID:PMC5753214 

 2/ Lines 52-54 

The previous paper was incorrectly stated. I have changed the content of the paper that “The cyclization of 2,3-oxidosqualene catalyzed by OSCs is a key step in the biosynthesis of triterpenoid saponins and sterols”.

3/ Line 56

I have mistakenly spelled before, and have already changed the word “Cholesterol” to “phytosterol”.

4/ Lines 87-89 (changed to lines 87-98)

A more detailed statement about the JA signaling mechanism is :

In the presence of JA-isoleucine, the SCFCOI1 complex forms, and JAZ proteins are degraded by the 26S proteasome. The ubiquitination of JAZ protein binds to substrate-specific SCFCOI1, and the degradation of JAZ protein by the 26S proteasome abolishes this interaction. The JAZ proteins contain a conserved TIFY motif within the ZIM domain that mediates homoand hetero-dimeric interactions between different JAZ proteins. The ZIM domain also functions to recruit transcriptional corepressors through the novel interactor of JAZ protein. The JAZ proteins contain Jas domain that is required for the interaction of both COI1 and a broad array of TFs. In the presence of JA or its bioactive derivatives, JAZ proteins are degraded and freeing TFs for expression of specific sets of JA-responsive genes, regulate enzymes involved in the biosynthesis of secondary metabolites, such as ginsenoside, artemisinin, and vinblastine. 

5/ Lines 104-107 (changed to lines 112-115)

The detailed treatment of plants is described below.

After 20 days of emergence, the leaves of plants were spraied with a mixture of different concentrations of NAA and 6-BA until there was liquid dripping at the edge of the blade. The mixture included: 1.0 mg/L NAA + 0.5 mg/L 6-BA (T1); 2.0 mg/L NAA + 1.0 mg/L 6-BA (T2); 4.0 mg/L NAA + 2.0 mg/L 6-BA (T3) and 8.0 mg/L NAA + 4.0 mg/L 6-BA (T4).

6/ Lines 126-128 (changed to lines 136-139)

The method for determining the chlorophyll (Chl. a, Chl. b) and carotenoid contents is expressed as follows:

The second pair of leaves from the top of plants (Control and T2 group) was harvested to measure the contents of chlorophyll a, b, and carotenoid. Fresh chopped leaves (0.1 g) added in 10 mL of 80% acetone (v/v) at room temperature in a dark environment for 24 h, and then measured the absorbance at 645 nm, 663 nm and 470 nm, respectively, for detecting chlorophyll a, b and carotenoids contents.

7/ Lines 252-254 (changed to lines 262-264)

Some genes indeed are part of different processes. Many of the genes in A. bidentata treated with NAA and 6-BA were affected because the various biological processes are complex and some genes are involved in other biological reaction process. Therefore, the percentage of different processes in the GO analysis is very high.

8/ Lines 297-299 (changed to lines 312-314)

About 28% of the 900 species in the Amaranthaceae estimated to occur in the family are C4 species, such as Gomphrena meyeniana, Amaranthus hypochondriacus.

The references is:

Sage RF, Sage TL, Pearcy RW, Borsch T. The taxonomic distribution of C4 photosynthesis in Amaranthaceae sensu stricto. Am J Bot. 2007; 94(12): 1992-2003. https://doi.org/10.3732/ajb.94.12.1992 PMID: 21636394

9/ Lines 313-314 (changed to lines 328-329)

After the modification, I have used bZIP and bHLH as abbreviation.

10/ Lines 346-347 (changed to lines 361-362)

The results are the data analysis we have done. The production of glycyrrhizin is oxidized at the C-11 position while oleanolic acid biosynthesis is at C-28 position oxidized. It may involved in ent-kaurene biosynthesis, because we observed that the ent-laurene synthase gene is up-regulated in the biosynthesis of gibberellin.

11/ Lines 347-349 (changed to lines 362-364)

The saponins currently isolated and identified from A. bidentata are all oleanane triterpenoid saponins. UGT can add sugar chains to oleanolic acid to form oleanane-type triterpenoid saponins, and UGT gene expression is up-regulated, which proves that more oleanane-type triterpenoid saponins are produced. From the side, it is indicated that the content of oleanolic acid is increased. So we analyzed the UGT slightly here.

12/ Remark Figure 4.

Your comments are meaningful, but the data we analyze does have some differences from your point of view. We also selected some genes for qRT-PCR validation, and the results were consistent with the expression pattern of RNA-seq assay. This result may be due to some differences in the expression of some enzyme genes in different species in MVA and MEP pathways.

Ginsenoside biosynthesis is part of our data analysis. In addition, ginsenosides have been reported in the achyranthes. Ginsenosides belong to the triterpenoids, one is the oleanane type pentacyclic triterpenoid saponin Ro, and the other two are the ginsengdiol type saponins (such as Rb1, Rb2, Rc, Rd, F2, Rg3, Rh2, etc.) and Ginseng triol saponins (such as Re, Rg1, Rg2, Rf, Rh1, etc.), both of which belong to the dammarane type tetracyclic triterpenoid saponins.

13/ Remark Figure 5

We used DEGs to analyze the gene data between samples, and analyzed the key enzymes involved in the synthesis of triterpenoid saponins. It was found that there were more differential genes involved in the synthesis of triterpenoid carbon skeleton, and 38 were co-expressed in control roots and treated leaves up-regulated CYP450s gene sequence. In order to further narrow the potential range of CYP450s involved in biosynthesis of oleanolic acid saponin, 25 up-regulated CYP450s (with sequence lengths > 1500 bp) were selected for phylogenetic analysis from A. bidentata treated leaves.

14/ Line 374 (changed to lines 389)

Your comment makes a lot of sense, we also consulted a lot of literature: more experiments have shown that MYC, MYB, WRKY and AP2/ERF transcription factors are involved in the regulation of terpene metabolism in different medicinal plants [1, 2]. For example, AaWRKY1, TSAR1, and TSAR2 are found to regulate the expression of HMGR genes in the MVA pathway in ArtemisiaannuaL. and Medicago truncatula. TSAR1 and TSAR2 are two transcription factors that are induced by jasmonic acid in the bHLH family [3, 4]. In the Medicago truncatula hairy roots, overexpression of TSAR1 and TSAR2 significantly increased the transcriptional expression levels of the MVA pathway gene and a series of genes involved in the synthesis of β-amyrin, such as HMGS, HMGRI, SS and β-AS [4, 5]. In our study, most AP2/ERF TFs, BHLHTFS and MYB TFs were up-regulated in treated leaves, and significantly increased the expression of triterpenoid biosynthesis genes in A. bidentata, especially HMGR, SS, SE and β-AS gene. Therefore, JA regulates oleanolic acid accumulation by freeing TFs and increasing expression of terpenoid biosynthesis genes in A. bidentata. These results were consistent with overexpression of TSARs to promote oleanane-type triterpene saponin in M. truncatula.

References:

[1] Afrin S, Jing-Jia H, Zhi-Yong L. JA-mediated transcriptional regulation of secondary metabolism inmedicinal plants[J]. Science Bulletin. 2015, 60(12): 1062-1072.

[2] Wasternack C, and Hause B. Jasmonates: biosynthesis, perception, signal transduction and action in plant stress response, growth and development. an update to the 2007 review in annals of botany[J]. Annals of Botany. 2013, 111(6): 1021.

[3] Patra B, Schluttenhofer C, Wu Y, et al. Transcriptional regulation of secondary metabolite biosynthesis in plants[J]. Biochimica Et Biophysica Acta. 2013, 1829(11): 1236-1247.

[4] Goossens A, Mertens J, Pollier J, et al. The bHLH Transcription Factors TSAR1 and TSAR2 Regulate Triterpene Saponin Biosynthesis in Medicago truncatula[J]. Plant Physiology. 2015, 170(1): 194.

[5] Mertens J, Moerkercke A V, Bossche R V, et al. Clade IVa Basic Helix -Loop -Helix Transcription Factors Form Part of a Conserved Jasmonate Signaling Circuit for the Regulation of Bioactive Plant Terpenoid Boynesis[J]. Plant and Cell Physiology. 2016, 57(12): 2564.

15/ Lines 389-391 (changed to lines 404-406)

We have already done it. (Li Jinting. Effects of Methyl Jasmonate on the Growth and Distribution of Main Medicinal Components in Achyranthes Bidentata Blume.)

Spraying 1.0mg/L MeJA could increase the content of oleanolic acid in A. bidentata.

16/ Lines 410-411 (changed to lines 425-426)

Spelling mistakes, change tertripenoid to triterpenoid.

Reviewer #2

Line 40

The language expression is not accurate and should be “Oleanolic acid, a pentacyclic triterpenoid, is an isoprenoid-derived compound ”. 

Oleanolic acid has been purified during the extraction process.

Preparation of oleanolic acid determination sample: 0.5 g sample was added with methanol 10 mL, ultrasonically extracted for 40 min, and then concentration using a rotary vacuum evaporator. To the concentrated samples 10 mL of 4 mol/L hydrochloric acid was added for hydrolyzing at 85°C for 1 h. After being allowed to cool, 10 mL of chloroform were added to the sample, followed by two countercurrent extractions at 60°C, each extraction being 15 min. The bottom liquid was harvested and evaporated under reduced pressure. Chromatography of pure methanol to a volume of 3 mL, mixed and filtered through a Millipore filter (0.22 μm), and determined by HPLC. The assay conditions were mobile phase: methanol‐water‐glacial acetic acid (90: 10: 0.05); flow rate: 0.9 mL/min; column temperature: 30°C; detection wavelength: 210 nm.

Line 103 (changed to lines 111)

Achyranthes bidentata is one of the famous Dao-di herbs in China, this species is primarily distributed in the Guhuaiqingfu area, located in Jiaozuo in Henan Province. So A. bidentata is also called ‘Huainiuxi’. Dao-di herbs mean that natural Chinese medicines that grow naturally under the influence of natural environment and social environment such as specific geography and climate, or are processed and cultivated with good quality and high curative effect and is synonymous with quality medicinal herbs in China. The quality of A. bidentata in Jiaozuo is better than in other regions, therefore, we choose to experiment in the natural environment.

Line 202 (changed to lines 212)

Here, it should be compared with the control, these characteristics showed a trend of increasing first and then decreasing. The reason we explore was lower concentrations of NAA and 6-BA promote the growth of A. bidentata while higher concentrations inhibited growth of A. bidentata. Therefor, it presenting this trend.

Line 204 (changed to lines 214-215)

We have replaced it with a more appropriate expression, using ‘growth parameters’ instead of ‘index’.

Line 257 (changed to lines 267-270)

I have improved the “For the pathway enrichment analysis, a total of 28 pathways were enriched (adjusted P < 0.05) with DEGs (Table 2)” to “To further identify metabolic or signal transduction pathways in which the DEGs are likely to be involved in promoting accumulation of oleanolic acid by JA regulation by NAA and 6-BA treatment in A. bidentata, and pathway enrichment analysis was performed using KEGG database. For the pathway enrichment analysis, a total of 28 pathways were enriched (adjusted P < 0.05) with DEGs (Table 2)”.

Line 318 (changed to lines 333)

We identified 22 key enzyme genes involved in the MVA pathway and 23 key enzyme genes involved in the MEP pathway. Comparing the expression in the roots and leaves of A. bidentata, it was found that the RPKM values of MCT, HDS and HDR genes were significantly higher than roots. The HMGS, HMGR, PMK and MDD genes in the MVA pathway are highly expressed in roots and low in leaves, which is completely opposite to the expression of key enzyme genes in the MEP pathway. These results are consistent with the expression of key enzyme genes in the MVA pathway and the MEP pathway in different vegetative organs of Astragalus membranaeus Bge. In addition, more documents revealed that protein of CMS, DXS, DXR and HDS enzymes are highly expressed in young roots and floral organs of Arabidopsis. These results clearly demonstrate that the MEP pathway synthesized in Plastid pathway prefers expressed in leaves, while the MVA pathway favors in roots. In root and leaf samples, it is clearly that chloroplasts are common in leaves and are relatively rare in roots. This difference may be due to the fact that the expression of key enzyme genes in the majority of MEP synthesis pathways is significantly higher in leaves than in roots and in the up-regulation of genes in roots in most MVA pathways. The expression patterns of key enzyme genes in different tissues in these two pathways indicated that the regulation of the biosynthetic pathways of IPP and DMAPP in different tissues of A. bidentata is parallel but separate.

---

## [Decision Letter · Decision Letter 1]

16 Oct 2019

PONE-D-19-21410R1

NAA and 6-BA promote accumulation of oleanolic acid by JA regulation in Achyranthes bidentata Bl. through de novo transcriptomics

PLOS ONE

Dear Mrs. Li,

Thank you for submitting your manuscript to PLOS ONE. After careful consideration, we feel that it has merit but does not fully meet PLOS ONE’s publication criteria as it currently stands. Therefore, we invite you to submit a revised version of the manuscript that addresses the points raised during the review process.

We would appreciate receiving your revised manuscript by Nov 30 2019 11:59PM. To enhance the reproducibility of your results, we recommend that if applicable you deposit your laboratory protocols in protocols.io, where a protocol can be assigned its own identifier (DOI) such that it can be cited independently in the future. For instructions see: http://journals.plos.org/plosone/s/submission-guidelines#loc-laboratory-protocols

We look forward to receiving your revised manuscript.

Kind regards,

Marie-Joelle Virolle, PhD

Academic Editor

PLOS ONE

Reviewers' comments:

Reviewer's Responses to Questions

**Comments to the Author**

1. If the authors have adequately addressed your comments raised in a previous round of review and you feel that this manuscript is now acceptable for publication, you may indicate that here to bypass the “Comments to the Author” section, enter your conflict of interest statement in the “Confidential to Editor” section, and submit your "Accept" recommendation.

Reviewer #1: (No Response)

Reviewer #2: (No Response)

2. Is the manuscript technically sound, and do the data support the conclusions?

Reviewer #1: Partly

Reviewer #2: Yes

3. Has the statistical analysis been performed appropriately and rigorously? 

Reviewer #1: Yes

Reviewer #2: Yes

4. Have the authors made all data underlying the findings in their manuscript fully available?

Reviewer #1: No

Reviewer #2: (No Response)

5. Is the manuscript presented in an intelligible fashion and written in standard English?

Reviewer #1: No

Reviewer #2: Yes

6. Review Comments to the Author

Reviewer #1: Several of my comments were addressed, however some points remain:

1/ I still don’t agree with the pathway presented in Figure 4. It is true that two distinct pathways, the cytosolic MVA pathway and the plastidial MEP pathway produce IPP and DMAPP. Exchange of IPP and DMAPP between the cytosol and the plastids can occur. However, downstream of IPP and DMAPP there is also a difference between the cytosol and the plastids. In the cytosol, 2 molecules of IPP and one molecule of DMAPP are used to make FPP by the FPPS enzyme. No GPPS enzyme is involved, and no GPP is produced, as is incorrectly shown in Figure 4. In the plastids, however, GPP is produced by GPPS, and that GPP can then be used for further downstream product formation (without FPP production). Please correct the pathway in Figure 4. For more information on the MVA-MEP pathways, see for instance: https://www.ncbi.nlm.nih.gov/pubmed/23451776

On the dammarene-type saponins (dammarenediol-derived) in this figure: if there are studies showing these compounds accumulate in Achyranthes spp then it is okay to include them in the figure; but in that case, refer to them in the paper. For now only oleanane-type (β-amyrin-derived) saponins are mentioned in the paper.

2/ Also, make sure to have a clear distinction between saponins containing an oleanolic acid aglycone and free oleanolic acid in your manuscript. In many plant species OA occurs as a free metabolite, whereas here it occurs as a saponin aglycone. When you measure OA, you carry out an acid hydrolysis to liberate the OA aglycones. In lines 37-39 you clearly state that OA occurs, not that OA occurs as a saponin aglycone. There is a big difference between the biological activity and in planta role of free OA (hydrophobic metabolite) and a saponin (amphipathic metabolite). To clarify throughout the manuscript.

3/ Line 54: correct typo to cycloartenol.

4/ Line 56: Little is known about genes involved in triterpenoid skeleton modification… This statement was true 10 years ago, but now plenty of genes are known to be involved, especially P450s.

5/ In the intro you point out that many phytohormones are used to increase the production of specialized metabolites. The most obvious treatment, jasmonates, is omitted in the first part, whereas later it is the one you focus on to explain the signaling mechanism. Also provide some examples of JA treatment, there are plenty of studies that show their effect on saponin metabolism.

6/ Line 113, typo: sprayed.

7/ The reads were submitted to the NCBI SRA database, but where can we find the sequence information of the assembled transcripts? The entire assembly – or at least the sequences that you use for phylogenetics or mention as candidate genes should be made publicly available.

8/ For the phylogenetic analysis, please provide the alignment file that was used to create the tree as SI. Why was a NJ tree opted for and not an ML tree?

9/ Figure 5. My comment was not addressed. In your figure, only 13 published P450s are included, whereas over 50 have been shown to be involved in triterpene (saponin) biosynthesis. Was there a reason to choose this specific set of P450s?

10/ Also my remark that there is not enough data to support the conclusion that JA regulates OA accumulation is not properly addressed. You provide a reference in your rebuttal, but this information was not used to update the manuscript.

Reviewer #2: I am not satisfied by the answer about methodology. I did not want to obtain a detailed description of extraction and quantification procedure as an answer to my remark. I consider it as an important part of a paper, and it should be inserted in a text. Particularly because the Authors menton in their manuscript free oleanolic acid and its saponins really in a very confusing way. For example:

Line 37 „Oleanolic acid is one of the major active phytochemical compounds in A. bidentata, and possesses hepatoprotective effects as well as anti-inflammatory, antioxidant, and anticancer activities”.

So finally –from this manuscript – the reader does not know if A. bidentata contains free oleanolic acid or its saponins? And it is important, because free oleanolic acid might have different bioactivity than its saponins.

Please precise:

Line 37. Oleanolic acid saponins are …

Line 130. Add the general description of extraction , ACID HYDROLYSIS and quantification procedure. It is important what you are measuring: saponins or free aglycon.

It should not be as detailed as in the answer to my question, but making the sequence of this procedure understandable for the reader.

Another problem not solved is the misunderstanding about the growth parameters of plants.

Line 211. After 20 days of treatment, the root length, fresh weight of root, dry weight of root, plant height, fresh weight of plant, and dry weight of plant had the same trend of first increasing then decreasing.

What do this description mean? „Fist increasing then decresing” could be understood as a tendency occurring in time, i.e. that the plant are first growing than are becoming smaller. And it is not true because this experiment is not made in time, it is a comparison of different treatments. So why „first increasing then decresing” if the Authors are mentioned independent experiments and NOT the process occurring in time? Only because they numbered the treatments as 1,2,3,4, and then what is occuring for the treatment 1 is „first” and for the treatment 4 – the last?

The figure illustrating these results is even more confusing because it is presented as a line of trend, and T can mean „time”. It would be better understandable if it would be a chart with separate points or bars.

The same problem is with the Figure 2.

According to the comment of the reviewer 1, the Authors changed „cholesterol” to „phytosterol”. It is better than previously, but still not really perfect solution, because „phytosterol” is not one compound, this name is used for a numerous group of plant sterols (e.g., sitosterol, stigmasterol, campesterol…etc.), and in every plant a mixture of at least several sterols is synthesized. So it should be writen „phytosterols” better than just „phytosterol”.

7. PLOS authors have the option to publish the peer review history of their article (what does this mean?). If published, this will include your full peer review and any attached files.

Reviewer #1: Yes: Jacob Pollier

Reviewer #2: No

---

## [Author Response · Author response to Decision Letter 1]

30 Nov 2019

We have tried our best to revise and improve the manuscript and made many changes in the manuscript according to the reviwers′good comments. We appreciate for Editors/Reviewers’ warm work earnestly, and hope that the corrections will meet with approval. 

Once again, thank you very much for your comments and suggestions. 

We look forward to your information about my revised papers and thank you for your good comments.

---

## [Decision Letter · Decision Letter 2]

18 Dec 2019

PONE-D-19-21410R2

NAA and 6-BA promote accumulation of oleanolic acid by JA regulation in Achyranthes bidentata Bl. through de novo transcriptomics

PLOS ONE

Dear Dr Jinting Li,

Thank you for submitting your revised manuscript to PLOS ONE. After careful consideration by two experts in the field, we feel that it has merit but does not fully meet PLOS ONE’s publication criteria as it currently stands. Therefore, we invite you to submit a novel revised version of the manuscript that addresses the rather minor points raised during the last review process.

We would appreciate receiving your revised manuscript by mid january. To enhance the reproducibility of your results, we recommend that if applicable you deposit your laboratory protocols in protocols.io, where a protocol can be assigned its own identifier (DOI) such that it can be cited independently in the future. For instructions see: http://journals.plos.org/plosone/s/submission-guidelines#loc-laboratory-protocols

We look forward to receiving your revised manuscript.

Kind regards,

Marie-Joelle Virolle, PhD

Academic Editor

PLOS ONE

Reviewers' comments:

Reviewer's Responses to Questions

**Comments to the Author**

1. If the authors have adequately addressed your comments raised in a previous round of review and you feel that this manuscript is now acceptable for publication, you may indicate that here to bypass the “Comments to the Author” section, enter your conflict of interest statement in the “Confidential to Editor” section, and submit your "Accept" recommendation.

Reviewer #1: (No Response)

Reviewer #2: (No Response)

2. Is the manuscript technically sound, and do the data support the conclusions?

Reviewer #1: Yes

Reviewer #2: Yes

3. Has the statistical analysis been performed appropriately and rigorously? 

Reviewer #1: Yes

Reviewer #2: Yes

4. Have the authors made all data underlying the findings in their manuscript fully available?

Reviewer #1: Yes

Reviewer #2: Yes

5. Is the manuscript presented in an intelligible fashion and written in standard English?

Reviewer #1: No

Reviewer #2: Yes

6. Review Comments to the Author

Reviewer #1: The authors now have addressed most of my comments, however, when they changed the manuscript a lot of typographical errors were introduced that should be corrected prior to publication of the manuscript. Some examples are (line numbers from track changes upload) are given below:

- Line 38: saponins is now plural, adjust "is one of the"

- Line 55-56: CAS = cycloartenol synthase, this has been changed but is now incorrect.

- Line 73: For example is repeated.

- Line 74: Treatment "of with" MeJA, remove "of"

- Line 83: remove one of the repeated commas.

- Line 88: "singly", please correct.

- Line 91: Here you use mmol L-1, before you used mg/L, please be consistent.

- Line 100: correct "homoand"

- Line 141: Add space before "Sample"

- Line 141: Add space between "methanol" and "was" and do something about the large (tab?) between "to" and "0.5"

- Line 142-150: re-write in proper English.

- Line 377: remove the space before the comma.

- Line 441: Only keep the numbered references.

- Line 486: italicize A. bindentata.

Overall the use of "S6 Table" or "S4 Fig" instead of "Table S6" of Fig. S4" sounds weird, please correct the order throughout the entire paper.

Reviewer #2: I have still some comments to the Authors and I feel that they did not understand some of my requests.

Line 37. Oleanane-type triterpenoid saponins is one of the major active phytochemical compounds in A. bidentata. This sentence is not correct grammatically (saponins are plural). It would be better to write „(..) saponins belong to the major active …

The description in the lines 225-229 is still not corrected and I see that the Authors does not understand my remark. The experiment is composed of 4 independent treatments, and the results cannot be commented as „first incresing then decreasing” only because the results for treatments T1 and T2 appear first (it means on the left side of the chart). The description „first.. then...” suggests the phenomenon occurring in time, and it is not the case in the presented experiment.

It would be better to omit this problem, writing:

The growth parameters of the plants after 20 days of treatment with various combinations of NAA and 6-BA are presented in Fig. 1. Lower concentrations of NAA and 6-BA (treatment T1 and T2) improved these parameters, while higher concentrations (particularly T4) exerted adverse effect.

The same remark for the description in lines 239-240.

Please definitively delete the sentence: „Oleanolic acid content rose at first and then fell.” The explanation which appear in the following sentences is totally sufficient and understandable.

7. PLOS authors have the option to publish the peer review history of their article (what does this mean?). If published, this will include your full peer review and any attached files.

Reviewer #1: Yes: Jacob Pollier

Reviewer #2: No

---

## [Author Response · Author response to Decision Letter 2]

6 Jan 2020

Dear editor and reviewers:

We have tried our best to revise and improve the manuscript and made many changes in the manuscript according to the reviwers′good comments. We appreciate for editor and reviewers’ warm work earnestly, and hope that the corrections will meet with approval. 

Once again, thank you very much for your comments and suggestions. 

We look forward to your information about my revised papers and thank you for your good comments.

---

## [Decision Letter · Decision Letter 3]

15 Jan 2020

PONE-D-19-21410R3

NAA and 6-BA promote accumulation of oleanolic acid by JA regulation in Achyranthes bidentata Bl. through de novo transcriptomics

PLOS ONE

Dear Jinting Li,

Thank you for submitting your revised manuscript to PLOS ONE. One of the two reviewers think that you did not fulfill his requests therefore I recommand again minor revision of your paper and invite you to submit a revised version of the manuscript that addresses the various points listed by this reviewer.

We would appreciate receiving your revised manuscript by end of january. To enhance the reproducibility of your results, we recommend that if applicable you deposit your laboratory protocols in protocols.io, where a protocol can be assigned its own identifier (DOI) such that it can be cited independently in the future. For instructions see: http://journals.plos.org/plosone/s/submission-guidelines#loc-laboratory-protocols

We look forward to receiving your revised manuscript.

Kind regards,

Marie-Joelle Virolle, PhD

Academic Editor

PLOS ONE

Reviewers' comments:

Reviewer's Responses to Questions

**Comments to the Author**

1. If the authors have adequately addressed your comments raised in a previous round of review and you feel that this manuscript is now acceptable for publication, you may indicate that here to bypass the “Comments to the Author” section, enter your conflict of interest statement in the “Confidential to Editor” section, and submit your "Accept" recommendation.

Reviewer #1: (No Response)

Reviewer #2: All comments have been addressed

2. Is the manuscript technically sound, and do the data support the conclusions?

Reviewer #1: Yes

Reviewer #2: Yes

3. Has the statistical analysis been performed appropriately and rigorously? 

Reviewer #1: Yes

Reviewer #2: Yes

4. Have the authors made all data underlying the findings in their manuscript fully available?

Reviewer #1: Yes

Reviewer #2: Yes

5. Is the manuscript presented in an intelligible fashion and written in standard English?

Reviewer #1: No

Reviewer #2: Yes

6. Review Comments to the Author

Reviewer #1: In my previous review i indicated that there were still many typographical and grammatical errors that should be corrected in addition to the ones indicated in my review. It seems the authors only corrected the errors i pointed to, hence, underneath a list containing more errors i spotted after carefully reading the paper.

1/ The title is not in standard English. I suggest to remove "through de novo transcriptomics" or reformulate the title so it is clear that de novo transcriptomics revealed the effect of NAA and 6-BA on JA signaling and OA biosynthesis.

2/ In the abstract (line 28) you still mention OA accumulation, whereas the plant accumulates saponins with an oleanolic acid backbone.

3/ Line 39, possesses should be possess.

4/ Line 52, add a space between diphosphate and synthase.

5/ Line 57, correct "Many are known about" to e.g. "A lot is known about" or even better, change the sentence to something like "Several genes involved in modification of the triterpenoid skeleton downstream of the cyclization step are known."

6/ Line 60, change "are assumed" to "were shown to".

7/ Line 63-66: There is confusion about "oleanane". Oleanane points to triterpenes / saponins with a β-amyrin backbone. Hence, CYP716A12 is not per se involved in the biosynthesis of oleanane-type triterpene aglycones, it is just involved in the modification of the oleanane backbone.

8/ Line 86. Split into two sentences, add a dot after A. bidentata roots and start a new sentence with "Soaking the seeds..."

9/ Lines 92 and 94: COI1 in SCFCOI1 should be superscript.

10/ Line 92-95; the JA signaling and degradation of JAZ proteins is not clear here. Rephrase to for instance "In the absence of JA-Ile, the JAZ repressor proteins bind to the MYC2 transcription factor, thereby blocking downstream JA response. In the presence of JA-Ile, COI1 binds to the JAZ proteins, which leads to their ubiquination by the SCFCOI1 complex and their subsequent degradation by the 26S proteasome. Degradation of the JAZ proteins releases MYC2, leading to transcriptional activation of the JA-responsive genes."

11/ Line 98, add the abbreviation (NINJA) after novel interactor of JAZ.

12/ Line 99, add "a" between contain and Jas to "The JAZ proteins contain a Jas domain..."

13/ Line 117, correct to "20 days after emerging, the leaves were sprayed with a mixture..."

14/ Line 140, rephrase to: "... hydrochloric acid was added, and the extract was hydrolyzed at 85..."

15/ Line 226, correct to "... the plants 20 days after treatment with..."

16/ Legend Fig 2, add a space between 6-BA. and Each.

17/ In the abstract (line 21-23) it is claimed that 62% of the upregulated genes encode proteins involved in hormone or terpenoid biosynthesis or TFs. Lines 267-269 only claim that of the 20896 DEGs, 13071 were upregulated. I cannot imagine all upregulated genes encode proteins involved in hormone or terpenoid biosynthesis or TFs. This should be corrected in the abstract.

18/ Correct to bHLH and bZIP in Table 4.

19/ Legend Figure 4: check the CMK abbreviation, remove "50"

20/ Line 419, correct to bHLH.

21/ Legend Figure 7: "through de novo sequencing"?

22/ Legend Table S6: italicize "A. bidentata"

Reviewer #2: I am satisfied by the improvements made by the Authors. I consider the manuscript as acceptable for publication.

7. PLOS authors have the option to publish the peer review history of their article (what does this mean?). If published, this will include your full peer review and any attached files.

Reviewer #1: Yes: Jacob Pollier

Reviewer #2: No

---

## [Author Response · Author response to Decision Letter 3]

6 Feb 2020

Dear editor and reviewers:

We have tried our best to revise and improve the manuscript and made many changes in the manuscript according to the reviwers′good comments. We appreciate for editor and reviewers’ warm work earnestly, and hope that the corrections will meet with approval. 

Once again, thank you very much for your comments and suggestions. 

We look forward to your information about my revised papers and thank you for your good comments.

Responds to reviewers:

Reviewer #1

1/ 

We have modified the title according to your comments. We have removed“through de novo transcriptomics”.

2/ (line 28, changed to line 27)

We have modified the manuscript according to your comments. We have changed “oleanolic acid” as “oleanane-type triterpenoid saponins”.

3/ ( line 39)

We have corrected as“possess”. 4/ (Line 52)

We have added a space between “diphosphate” and “synthase”.

5/ (line 57)

We have modified the manuscript according to your comments. 

6/ (line 60)

We have changed "are assumed" to "were shown to".

7/ ( line 63-66, changed to line 65)

We have modified the manuscript according to your comments, and changed as “oleanane backbone”.

8/ (line 86, changed to line 85)

We have modified the manuscript according to your comments. 

9/ (line 92-95, changed to line 90-96)

We have modified the manuscript according to your comments. 

10/ (line 98, changed to line 99)

We have added the abbreviation “NINJA” after “novel interactor of JAZ”.

11/ (line 99)

We have added "a" between “contain” and “Jas”. 

12/(line 117, changed to line 118-119)

We have modified the manuscript according to your comments.

13/ (line 140)

We have modified the manuscript according to your comments.

14/ (line 226)

We have corrected to "... the plants 20 days after treatment with..."according to your comments.

15/ ( Legend Fig 2)

We have added a space between “6-BA.” and “Each”.

16/ (line 21-23, changed to line 20-23)

We have corrected it in the abstract.

17/ (Table 4)

We have corrected to “bHLH” and “bZIP”.

18/ ( Legend Figure 4)

We have rechecked the “CMK’’ abbreviation, and no errors found; and removed “50”.

19/ ( Line 419)

We have corrected to “bHLH”.

20/ ( Legend Figure 7)

We have deleted "through de novo sequencing", and changed to “Fig 7. NAA and 6-BA promote accumulation of oleanolic acid by JA regulation in A. bidentata.”

21/ (Legend Table S6)

We have changed to "A. bidentata".

Reviewer #2

Thank you very much for your comments on the manuscript. Your comments have made the manuscript more clear and reasonable. Thank you again!

---

## [Decision Letter · Decision Letter 4]

10 Feb 2020

NAA and 6-BA promote accumulation of oleanolic acid by JA regulation in Achyranthes bidentata Bl.

PONE-D-19-21410R4

Dear Dr. Li,

We are pleased to inform you that your manuscript has been judged scientifically suitable for publication and will be formally accepted for publication once it complies with all outstanding technical requirements.

With kind regards,

Marie-Joelle Virolle, PhD

Academic Editor

PLOS ONE

Additional Editor Comments (optional):

Reviewers' comments:

Reviewer's Responses to Questions

**Comments to the Author**

1. If the authors have adequately addressed your comments raised in a previous round of review and you feel that this manuscript is now acceptable for publication, you may indicate that here to bypass the “Comments to the Author” section, enter your conflict of interest statement in the “Confidential to Editor” section, and submit your "Accept" recommendation.

Reviewer #1: All comments have been addressed

Reviewer #2: All comments have been addressed

2. Is the manuscript technically sound, and do the data support the conclusions?

Reviewer #1: Yes

Reviewer #2: Yes

3. Has the statistical analysis been performed appropriately and rigorously? 

Reviewer #1: Yes

Reviewer #2: Yes

4. Have the authors made all data underlying the findings in their manuscript fully available?

Reviewer #1: Yes

Reviewer #2: Yes

5. Is the manuscript presented in an intelligible fashion and written in standard English?

Reviewer #1: Yes

Reviewer #2: Yes

6. Review Comments to the Author

Reviewer #1: (No Response)

Reviewer #2: Please correct the position 25 in the list of cited publications. It was added during the revision process and it is written with mistakes. The name of the first author is Alsoufi, the other Authors' names are Pączkowski, Szakiel, Długosz. It should be also added officinalis after Calendula.

7. PLOS authors have the option to publish the peer review history of their article (what does this mean?). If published, this will include your full peer review and any attached files.

Reviewer #1: Yes: Jacob Pollier

Reviewer #2: No

---

## [Editor Report · Acceptance letter]

13 Feb 2020

PONE-D-19-21410R4 

NAA and 6-BA promote accumulation of oleanolic acid by JA regulation in *Achyranthes bidentata* Bl.

Dear Dr. Li:

I am pleased to inform you that your manuscript has been deemed suitable for publication in PLOS ONE. Congratulations! Your manuscript is now with our production department. 

With kind regards,

on behalf of

Dr. Marie-Joelle Virolle 

Academic Editor

PLOS ONE